# NR-SAFE: a randomized, double-blind safety trial of high dose nicotinamide riboside in Parkinson's disease

Haakon Berven[1,2,3], Simon Kverneng[1,2,3], Erika Sheard[1], Mona Søgnen[1], Solveig Amdahl Af Geijerstam[1], Kristoffer Haugarvoll [1], Geir-Olve Skeie[1,2], Christian Dölle [1,2,3] ✉ & Charalampos Tzoulis [1,2,3] ✉

Nicotinamide adenine dinucleotide (NAD) replenishment therapy using nicotinamide riboside (NR) shows promise for Parkinson's disease (PD) and other neurodegenerative disorders. However, the optimal dose of NR remains unknown, and doses exceeding 2000 mg daily have not been tested in humans. To evaluate the safety of high-dose NR therapy, we conducted a single-center, randomized, placebo-controlled, double-blind, phase I trial on 20 individuals with PD, randomized 1:1 on NR 1500 mg twice daily ($n = 10$) or placebo ($n = 10$) for four weeks. The trial was conducted at the Department of Neurology, Haukeland University Hospital, Bergen, Norway. The primary outcome was safety, defined as the frequency of moderate and severe adverse events. Secondary outcomes were tolerability defined as frequency of mild adverse events, change in the whole blood and urine NAD metabolome, and change in the clinical severity of PD, measured by MDS-UPDRS. All 20 participants completed the trial. The trial met all prespecified outcomes. NR therapy was well tolerated with no moderate or severe adverse events, and no significant difference in mild adverse events. NR therapy was associated with clinical improvement of total MDS-UPDRS scores. However, this change was also associated with a shorter interval since the last levodopa dose. NR greatly augmented the blood NAD metabolome with up to 5-fold increase in blood NAD$^+$ levels. While NR-recipients exhibited a slight initial rise in serum homocysteine levels, the integrity of the methyl donor pool remained intact. Our results support extending the dose range of NR in phase II clinical trials to 3000 mg per day, with appropriate safety monitoring. Clinicaltrials.gov identifier: NCT05344404.

Parkinson's disease (PD) affects 1–2% of the population over 65 years, and is one of the fastest growing neurological disorders worldwide[1,2]. Current treatments for PD provide partial and temporary symptomatic relief for motor symptoms, but make no impact on the rate of neurodegeneration and disease progression[1,3,4].

Increasing cellular levels of nicotinamide adenine dinucleotide (NAD) has been proposed as a potential neuroprotective therapeutic

[1]Neuro-SysMed, Department of Neurology, Haukeland University Hospital, 5021 Bergen, Norway. [2]Department of Clinical Medicine, University of Bergen, Pb 7804, 5020 Bergen, Norway. [3]K.G. Jebsen Center for Translational Research in Parkinson's disease, University of Bergen, Pb 7804, 5020 Bergen, Norway. ✉ e-mail: Christian.Doelle@uib.no; charalampos.tzoulis@uib.no

strategy for neurodegeneration[5]. NAD, which constantly shuttles between its oxidized ($NAD^+$) and reduced (NADH) state, is a coenzyme required for metabolic reduction-oxidation (redox) reactions, including mitochondrial respiration, which are integral to cellular energy metabolism. In addition, $NAD^+$ is substrate for a number of vital non-redox reactions that continuously degrade the molecule, requiring constant replenishment[6,7].

NAD levels decline with age, possibly due to multiple and only partially understood processes, and this has been proposed to contribute to age-related diseases, including parkinsonisms[5-7]. Conversely, increasing the NAD replenishment rate via supplementation of precursors has shown beneficial effects on life- and healthspan in multiple animal models, and evidence of neuroprotection in models of parkinsonism and other neurodegenerative diseases[7-10]. NAD-replenishment may target multiple processes associated with PD and other neurodegenerative disorders, including mitochondrial function and energy metabolism[11,12], DNA-repair[13-15], epigenetic regulation[5,15,16], neuroinflammation[5,15,17,18], mitophagy[5,15], and lysosomal and proteasomal pathways[18]. In this manner, NAD-replenishment may increase neuronal resilience, shielding neurons against multiple forms of disease-associated stress[5,15].

In the NADPARK study, a recent randomized double-blind clinical trial of NAD replenishment therapy with nicotinamide riboside (NR) in PD, we showed that oral intake of 1000 mg NR daily for 30 days significantly augments cerebral NAD levels[19]. This was, in turn, associated with altered cerebral metabolism and a mild trend for clinical improvement[18]. Furthermore, NR was associated with transcriptional upregulation of processes related to mitochondrial, lysosomal, and proteasomal function in blood cells and/or skeletal muscle[18].

These findings suggested that NR may hold promise as a neuroprotective therapy for PD, which warrants further clinical investigation. An important knowledge gap in that regard is that the safety range and optimal dose of NR therapy for PD remain undetermined. In the setting of a potential neuroprotective intervention, it is critical that the given dose is sufficient to achieve the desired effects. While the NADPARK trial showed a robust neurometabolic response to NR, the effect size was variable, and not all NR-recipients responded[18]. It is possible that the neurometabolic effects of NR are dose-dependent and that higher doses, which remain unexplored to date, may prove more beneficial in terms of both effect size and proportion of responders.

In preclinical studies on rats, adverse effects appeared with doses beyond 1000 mg/kg/day, which corresponds to approximately 9700 mg/day for a 60 kg adult human[20]. Side effects included increased levels of neutrophils and triglycerides in blood, as well as weight loss and decreased organ weight of liver and kidneys[21,22]. Moreover, a dose of 3000 mg/kg/day (corresponding to ca. 29 g/day for a 60 kg adult human) resulted in abnormal function in the liver, kidneys, testes, epididymides and ovaries, with signs of chronic progressive nephropathy and centrilobular hepatocellular hypertrophy[21].

Most clinical studies with NR in adult humans to date have used up to 1000 mg daily[23-33], while three studies used 2000 mg daily[34-36]. None of these reported any evidence of toxicity or severe adverse effects. Long term treatment for 5 months with a daily dose of 1000 mg NR also did not reveal any adverse effects[33].

To determine the safety of high-dose NR therapy, we performed a double-blind, randomized phase I trial of 3000 mg NR daily in individuals with PD. The primary outcome of the study was the incidence of treatment-associated moderate and severe adverse events (AEs). Secondary outcomes were the between-group difference in treatment-associated mild AEs, change in the clinical severity of PD, measured by the Movement Disorder Society Unified Parkinson's Disease Rating Scale (MDS-UPDRS)[37], and changes of the NAD metabolome in blood and urine. Exploratory outcomes included the between-group difference in the change of serum homocysteine levels and of fasting blood glucose and serum insulin levels.

We show that oral NR treatment at a dose of 3000 mg daily for a 4-week period is well tolerated without moderate or severe adverse events, induces a pronounced augmentation of the NAD metabolome, and may be associated with clinical symptomatic improvement in PD. These findings support the possibility of extending the dose range of NR up to 3000 mg in clinical trials, provided adequate safety monitoring. However, the long-term safety of this dose has not yet been established.

## Results
### Study population
In total, 26 individuals with PD were screened, and 20 eligible participants were enrolled, all of whom completed the study (Fig. 1). There were no significant differences in sex, age, body mass index (BMI), time since diagnosis of PD, baseline MDS-UPDRS, or Hoehn and Yahr stage, between the NR and placebo group (Table 1). While not statistically significant, we noted that the NR group was on average slightly younger, and had a slightly lower BMI, fewer years since diagnosis, and higher baseline MDS-UPDRS (Table 1). Drug compliance was similar in both groups (NR: 95.6 ± 2.45%; Placebo: 94.4 ± 3.27%, paired Wilcoxon test: $p = 0.52$). There were no significant differences in the frequency of co-morbid medical conditions (Supplementary Data 1).

Samples from all individuals were included in the analyses of laboratory values, including those with abnormal values at baseline. One participant in the placebo group did not fast at any visit due to insulin-treated diabetes mellitus, and one participant in the NR group did not fast at baseline due to omission. These two individuals were omitted from analyses of values depending on fasting samples (i.e., insulin, lipids, and glucose). Two participants did not use levodopa and were therefore excluded in the analysis of the time since levodopa. Participants did not fast on visits other than baseline (visit 1, V1) and last visit (visit 7, V7), because this interfered too much with their daily function. One participant in the NR group was removed from the analysis of urea, sodium, potassium and creatinine due to data entry error. Pregnancy screening, with serum human chorionic gonadotrophin (HCG), was negative at baseline for all female participants. The groups were similar at baseline with regards to demographics, clinical safety tests, MDS-UPDRS scores and metabolomic parameters (Table 1, Supplementary Data 1–6).

### High dose NR treatment is not associated with adverse events
Forty-two adverse events (AEs) were observed overall, 25 of which were in the NR group and 17 in the placebo group. 9/10 participants in the NR group and 8/10 participants in the placebo group experienced at least one AE (Table 2, Supplementary Data 2). All AEs were graded as mild, and there was no significant difference in the frequency of adverse events between both groups. The most common adverse events observed in the NR group were classified as extrapyramidal disorder (number of events/number of affected individuals, $n = 3/3$), headache ($n = 3/3$), tremor ($n = 2/2$), muscle cramp ($n = 2/1$), fatigue ($n = 2/2$), nausea ($n = 2/2$) and dyspepsia ($n = 2/2$). All cases of dyspepsia and nausea were resolved at the end of the follow up period (Supplementary Data 2). One event each of fatigue, headache and muscle cramps was registered as ongoing at the end of the study. Regarding extrapyramidal disorders, one participant reported increased dyskinesia during the study causing the participant to reduce their daily levodopa dose, while another participant reported increased rigidity after treatment cessation with NR, and the third participant reduced their levodopa dosage during the study due to reduced subjective need of treatment. Regarding tremor, one participant experienced increased tremor during the study, whereas another experienced increased tremor after cessation of the study drug. One case of non-painful maculo-papular rash was seen in the NR group. This

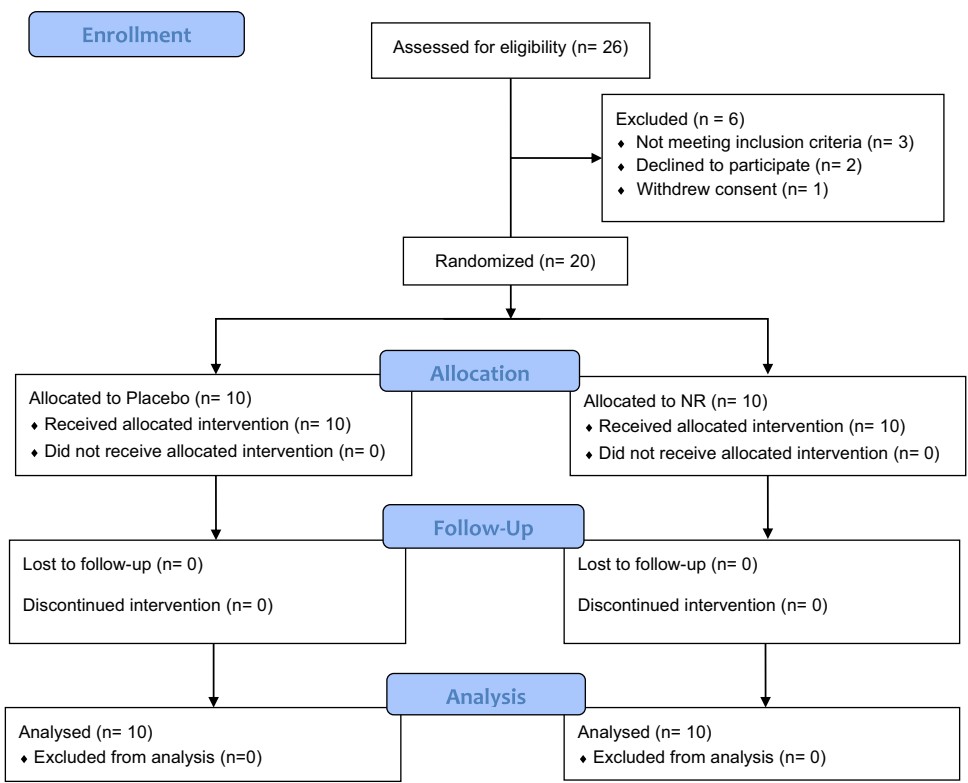

**Fig. 1 | Participant screening and randomization.** 26 patients with PD were screened for inclusion and 20 enrolled in the study. All randomized participants completed the study.

participant had recently been treated by their general practitioner for the same rash and, thus, a causal relationship was deemed unlikely. Importantly, no painful flushing was reported, which can occur with nicotinic acid (NA) supplementation as NAD precursor at dosages above 50–100 mg[38].

No significant changes in systolic or diastolic blood pressure, pulse or weight were seen between visit 1 and visit 7 (i.e., Δ = V7 − V1) in either group, nor were there significant between-group differences in the mean change for each variable (Table 3). Electrocardiography (ECG) revealed no changes of clinical significance (Supplementary Data 2). Two patients in the NR group developed asymptomatic

bradycardia (58 beats/min and 46 beats/min). Except from a small increase in homocysteine in the NR group (see below), no abnormal changes in clinical laboratory values were seen (Supplementary Data 3).

**High dose NR treatment is associated with clinical improvement**
The NR group, but not the placebo group, showed a statistically significant decrease in the total MDS-UPDRS (I-IV) score between V1 and V7 (NR: V1: 51 ± 21.28 vs V7: 40.3 ± 17.1; mean change −10.7 ± 9.94; paired t-test, p = 0.007; Placebo: V1: 41.8 ± 16.46 vs V7: 41.8 ± 22.23; mean change 0 ± 9.59; paired t-test, p = 1; Table 4 and Fig. 2a–e). The MDS-UPDRS change in the NR-group was significantly higher compared to the placebo (t-test, p = 0.024). To determine which symptoms had improved in the NR group, we performed a post hoc explorative analysis of the subparts of MDS-UPDRS (I-IV). The change in the NR group was primarily driven by MDS-UPDRS part III (V1: 29.7 ± 12.85 vs V7: 22.7 ± 7.55; paired t-test, p = 0.0214), which measures the severity of motor symptoms, and to a minor extent by part I (NR: −2.20 ± 3.52, t-test, p = 0.079), which measures non-motor experiences of daily living. The statistical significance for the changes in the subparts of MDS-UPDRS did not survive multiple testing correction (Table 4 and Fig. 2a–e).

We noted that the "mean time since the last levodopa dose" was shorter on day 28 compared to baseline in the NR group (−49 ± 71.92 min) and longer in the placebo group (+28.33 ± 105.58 min). Levodopa strongly influences motor symptoms in PD, and the time of intake commonly varies in response to symptom severity[39]. While all participants were assessed in the ON-state and reported no wearing-off, one cannot exclude the possibility that the observed change in MDS-UPDRS was, at least in part, driven by the difference in levodopa dosing intervals. To investigate this further, we assessed a selection of 8 participants per group in whom the difference in the intervals since last levodopa dose was less prominent between the NR (−25.85 ± 62.86 min) and placebo

## Table 1 | Demographic and clinical parameters at baseline (V1)

| Parameter | Placebo (±sd) n = 10 | NR (±sd) n = 10 | p[a] |
|---|---|---|---|
| Sex (female/male) | 3/7 | 2/8 | 1 |
| Age (years) | 65.5 ± 9 | 61.4 ± 9.28 | 0.33 |
| Height (cm) | 173.9 ± 8.01 | 177 ± 7.75 | 0.39 |
| Weight (kg) | 81.19 ± 10.90 | 79.22 ± 11.25 | 0.69 |
| Time since PD diagnosis (years) | 6.9 ± 4.15 | 5.4 ± 2.91 | 0.37 |
| BMI (kg/m²) | 26.79 ± 2.53 | 25.24 ± 2.86 | 0.22 |
| Hoehn & Yahr >2 | 0 | 2 | 0.47 |
| Total MDS-UPDRS | 41.8 ± 16.46 | 51 ± 21.28 | 0.29 |
| MDS-UPDRS Part I | 6.6 ± 3.92 | 7.8 ± 5.37 | 0.57 |
| MDS-UPDRS Part II | 8.2 ± 2.74 | 9.4 ± 6.25 | 0.58 |
| MDS-UPDRS Part III | 24.7 ± 11.95 | 29.7 ± 12.85 | 0.37 |
| MDS-UPDRS Part IV | 2.3 ± 2.94 | 4.1 ± 4.88 | 0.33 |

P-values not corrected for multiple testing. Source data are provided as a Source Data file.
[a]One-sided Fisher's exact test was used for comparing sex ratios and number of participants with Hoehn & Yahr scale score above 2. The remaining parameters were calculated using independent two-tailed t-test for normally distributed data and independent two-tailed Wilcoxon test for non-normally distributed data.

**Table 2 | Adverse events**

| Adverse event | NR ($n = 10$) | Placebo ($n = 10$) | Causal relationship to study drug ($n$) | $p$[j] | CTCAE[i] grading Grade 1–5 ($n$) |
|---|---|---|---|---|---|
| Headache | 3 | 0 | Possible (1) Unlikely (2) | 0.21 | Grade 1 (3) |
| Dyspepsia | 2 | 1 | Possible (3) | 1 | Grade 1 (3) |
| Nausea | 2 | 0 | Possible (2) | 0.47 | Grade 1 (2) |
| Dizziness | 1 | 1 | Possible (2) | 1 | Grade 1 (2) |
| Fatigue | 2 | 1 | Unlikely (2) Possible (1) | 1 | Grade 1 (3) |
| Dry mouth | 1 | 0 | Possible (1) | 1 | Grade 1 (1) |
| Abdominal pain | 1 | 3 | Probable (1) Possible (2) Unlikely (1) | 0.58 | Grade 1 (4) |
| Rash maculo-papular | 1 | 1 | Unlikely (2) | 1 | Grade 1 (2) |
| Insomnia | 1 | 0 | Unlikely (1) | 1 | Grade 1 (1) |
| Upper respiratory infection | 1 | 1 | Unlikely (2) | 1 | Grade 1 (2) |
| Urinary frequency | 1 | 0 | Unlikely (1) | 1 | Grade 1 (1) |
| Muscle cramp[a,b] | 2 | 0 | Possible (1) Unlikely (1) | 0.47 | Grade 1 (2) |
| Chest pain - cardiac | 0 | 1 | Unlikely (1) | 1 | Grade 1 (1) |
| Tremor[c] | 2 | 2 | Possible (4) | 1 | Grade 1 (4) |
| Localized edema | 0 | 1 | Unlikely (1) | 1 | Grade 1 (1) |
| Extrapyramidal disorder[d,e,f,g] | 3 | 1 | Possible (4) | 0.58 | Grade 1 (4) |
| Gastrointestinal disorders - other - abdominal gas discomfort | 1 | 1 | Possible (1) Probable (1) | 1 | Grade 1 (2) |
| Gastrointestinal disorders - other - increased salivation | 0 | 1 | Possible (1) | 1 | Grade 1 (1) |
| Metabolism and nutrition disorders - Decrease in blood glucose within normal range[h] | 0 | 1 | Possible (1) | 1 | Grade 1 (1) |
| Injury, poisoning and procedural complications - Contusion of ankle | 1 | 0 | Unrelated (1) | 1 | Grade 1 (1) |
| Injury, poisoning and procedural complications - Mild head trauma | 0 | 1 | Unlikely (1) | 1 | Grade 1 (1) |

$P$-values not corrected for multiple testing. Source data are provided as a Source Data file.
[a]Multiple adverse events reported by one subject.
[b]Former medical history of dystonia. Increased muscle cramps one week after cessation of the study drug.
[c]One patient in each treatment group reported increased tremor in the first week after cessation of study drug. The other two reported increased tremor during study drug administration.
[d]One participant (NR) reported increased dyskinesia during the study, causing the participant to auto-reduce levodopa dosage.
[e]One participant (placebo) reported decreased dyskinesia during study.
[f]One participant (NR) reported increased rigidity one week after study drug cessation.
[g]One participant (NR) auto-reduced their levodopa dose during the study due to reduced subjective need of medication.
[h]Caused diabetic participant to auto-reduce insulin dosage.
[i]Common Terminology Criteria for Adverse Events v5.0.
[j]One-sided Fisher's exact test.

($-2.85 \pm 98.77$ min) groups. This analysis revealed a still significant decrease in total MDS-UPDRS in the NR group (V1: $46.62 \pm 17.75$ vs V7: $35.75 \pm 11.74$; mean change $-10.87 \pm 11.15$; paired t-test, $p = 0.028$) but not in the placebo group (Fig. 2f, Supplementary Data 4). Nevertheless, the NR group still has a slight advantage in form of a slightly shorter interval since last levodopa dose and, given the small sample size and large interindividual variation, the possibility that the observed effect is attributed to levodopa cannot be confidently excluded.

**High dose NR treatment augments the NAD metabolome**
Next, we explored the effect of 3000 mg NR daily on the NAD metabolome in blood and urine. First, flash frozen whole blood samples were analyzed for $NAD^+$, NADH, $NADP^+$, NADPH and the reduced (GSH) and oxidized (GSSG) forms of glutathione, using the NADMed assay (see methods). The NR group showed a marked increase in the levels of $NAD^+$ and NADH, which resulted in an overall elevated $NAD^+$/NADH ratio (Fig. 3a–d). Interestingly, we also observed an increase in $NADP^+$ and total NADP levels, but not NADPH, and a resulting trend towards increased $NADP^+$/NADPH ratio ($p = 0.056$, Fig. 3e–h). In contrast, GSH and GSSG remained unchanged in both groups (Fig. 3i–l, Supplementary Data 5).

Blood samples were also subjected to targeted liquid chromatography-mass spectrometry (LC-MS) metabolomics analyses, with a focus on NAD-related metabolites, as well as untargeted metabolomics analyses (see methods). The NR group exhibited a significant increase in whole blood $NAD^+$ and $NADP^+$ levels, confirming the findings of the NADMed assay (Fig. 4e, f). Multiple metabolites involved in NAD biosynthesis and metabolism were increased as well, including nicotinamide (Nam) and its breakdown products nicotinamide N-oxide (Nam N-oxide), 1-methyl nicotinamide (Me-Nam), and N1-methyl-2-pyridone-5-carboxamide (Me-2-PY; Fig. 4g–j). Nicotinic acid adenine dinucleotide (NAAD) was also elevated as consistently observed in NR-supplementation studies[26,28,33]. Interestingly, we also detected elevated levels of nicotinamide mononucleotide (NMN; Fig. 4c). Furthermore, the NR group showed an increase in the NAD-derived signaling molecule ADP-ribose (ADPR) (Fig. 4q). In contrast, no significant changes were seen in the placebo group for any of the metabolites. The mean of the individual changes ($\Delta$ = V7−V1) in each of these metabolites was also significantly different between the NR and placebo groups, with the exception of $NADP^+$, which, however, showed a strong trend after multiple testing correction ($p = 0.057$; Supplementary Data 6). ATP and GTP levels were unchanged in both groups, as were ADP, AMP,

**Table 3 | Vital signs and body metrics**

| Parameter | Visit | Placebo (±sd) n = 10 | NR (±sd) n = 10 | pᵃ |
|---|---|---|---|---|
| Diastolic blood pressure (mmHg) | V1 | 78.30 ± 11.78 | 74.70 ± 8.49 | |
| | V7 | 73.10 ± 4.30 | 76.20 ± 6.81 | |
| | Δ | −5.20 ± 12.58 | 1.50 ± 8.31 | 0.34 |
| | pᵇ | 0.31 | 1 | |
| Pulse (min⁻¹) | V1 | 67.80 ± 7.71 | 64.20 ± 14.67 | |
| | V7 | 71.40 ± 9.91 | 61.10 ± 12.95 | |
| | Δ | 3.60 ± 10.18 | −3.10 ± 9.48 | 0.17 |
| | pᵇ | 0.35 | 0.52 | |
| Systolic blood pressure (mmHg) | V1 | 131.40 ± 21.46 | 122.10 ± 12.52 | |
| | V7 | 121.60 ± 7.97 | 122.20 ± 7.49 | |
| | Δ | −9.80 ± 23.25 | 0.10 ± 8.29 | 0.23 |
| | pᵇ | 0.21 | 0.97 | |
| Weight (kg)ᶜ | V1 | 81.19 ± 10.90 | 79.38 ± 11.92 | |
| | V7 | 80.73 ± 10.79 | 79.38 ± 11.79 | |
| | Δ | −0.46 ± 0.89 | 0.00 ± 0.58 | 0.20 |
| | pᵇ | 0.14 | 1 | |

For comparison within one group, paired two-tailed t-test or two-tailed Wilcoxon test was used. For comparison between treatment groups, independent two-tailed t-test or two-tailed Wilcoxon test was used. Non-normally distributed data was analyzed by two-tailed Wilcoxon test. *P*-values not corrected for multiple testing. Source data are provided as a Source Data file.
ᵃ*P*-values are comparisons of Δ values between the NR and placebo group.
ᵇ*P*-value are comparisons between V7 and V1 in each treatment group.
ᶜPlacebo *n* = 10, NR *n* = 9.

**Table 4 | MDS-UPDRS and time since levodopa**

| Parameter | Visit | Placebo (±sd, n = 10) | NR (±sd, n = 10) | p-valueᵃ | p-adjᶜ |
|---|---|---|---|---|---|
| Total MDS-UPDRS | V1 | 41.80 ± 16.46 | 51.00 ± 21.28 | | |
| | V7 | 41.80 ± 22.23 | 40.30 ± 17.10 | | |
| | Δ | 0.00 ± 9.59 | −10.70 ± 9.94 | 0.02 | NA |
| | *p*-valueᵈ | 1 | 0.007 | | |
| | *p*-adjᶜ | NA | NA | | |
| MDS-UPDRS Part I | V1 | 6.60 ± 3.92 | 7.80 ± 5.37 | | |
| | V7 | 6.30 ± 4.99 | 5.60 ± 5.01 | | |
| | Δ | −0.30 ± 2.26 | −2.20 ± 3.52 | 0.17 | 0.22 |
| | *p*-valueᵈ | 0.68 | 0.07 | | |
| | *p*-adjᶜ | 0.91 | 0.15 | | |
| MDS-UPDRS Part II | V1 | 8.20 ± 2.74 | 9.40 ± 6.25 | | |
| | V7 | 7.50 ± 4.17 | 8.80 ± 7.58 | | |
| | Δ | −0.70 ± 3.26 | −0.60 ± 2.36 | 0.93 | 0.93 |
| | *p*-valueᵈ | 0.51 | 0.44 | | |
| | *p*-adjᶜ | 0.91 | 0.44 | | |
| MDS-UPDRS Part III | V1 | 24.70 ± 11.95 | 29.70 ± 12.85 | | |
| | V7 | 24.60 ± 16.46 | 22.70 ± 7.55 | | |
| | Δ | −0.10 ± 6.79 | −7.00 ± 7.98 | 0.05 | 0.17 |
| | *p*-valueᵈ | 0.96 | 0.02 | | |
| | *p*-adjᶜ | 0.96 | 0.08 | | |
| MDS-UPDRS Part IV | V1 | 2.30 ± 2.94 | 4.10 ± 4.88 | | |
| | V7 | 3.40 ± 3.59 | 3.20 ± 3.15 | | |
| | Δ | 1.10 ± 2.33 | −0.90 ± 2.60 | 0.08 | 0.17 |
| | *p*-valueᵈ | 0.16 | 0.30 | | |
| | *p*-adjᶜ | 0.67 | 0.40 | | |
| Time since levodopa (minutes)ᵇ | V1 | 89.00 ± 79.25 | 154.33 ± 88.51 | | |
| | V7 | 117.33 ± 31.71 | 105.33 ± 61.60 | | |
| | Δ | 28.33 ± 105.58 | −49.00 ± 71.92 | 0.09 | NA |
| | *p*-valueᵈ | 0.44 | 0.07 | | |
| | *p*-adjᶜ | NA | NA | | |

For comparison within one group, paired two-tailed t-test or two-tailed Wilcoxon test was used. For comparison between treatment groups, independent two-tailed t-test or two-tailed Wilcoxon test was used. Non-normally distributed data was analyzed by two-tailed Wilcoxon test. Source data are provided as a Source Data file.
ᵃ*P*-values are comparisons of Δ values between the NR and placebo group.
ᵇPlacebo *n* = 9, NR *n* = 9. One individual in each group did not use levodopa.
ᶜP-adj denotes adjusted p-values via the Benjamini–Hochberg method.
ᵈ*P*-values are comparisons between V7 and V1 in each treatment group.

adenosine and GDP, similar to previous observations (Fig. 4k–p)[18]. NR itself did not increase in the NR group ($p = 0.135$), but the mean of the individual changes between the placebo and NR groups was significantly different ($p = 0.029$, Supplementary Data 6). In urine, the NR group exhibited an increase in several NAD-related metabolites, including NAR, Nam and the Nam breakdown products Me-Nam, Me-2-PY and Nam N-oxide, while no significant changes were seen in the placebo group (Fig. 4s–y). Also here, the mean of the individual changes ($Δ = V7–V1$) in each metabolite was significantly different between the NR and placebo groups (Supplementary Data 6). Similar to blood, urine NR itself did not change significantly in the NR group ($p = 0.061$), but again, the difference of the means of the individual changes between placebo and NR groups reached significance ($p = 0.03$, Supplementary Data 6). Coenzyme A (CoA), Acetyl-CoA and O-acetyl-ADP-ribose were also tested, but were below the limit of detection in the applied method.

## High dose NR treatment is not associated with depletion of the methyl group pool

One re-occurring concern in the literature is that NAD precursor supplementation may lead to methyl group depletion due to an increased elimination of Nam by methylation[40]. To investigate this possibility, we assessed related metabolites, including homocysteine (HCy), S-adenosyl-methionine (SAM) and S-adenosyl-homocysteine (SAH), as well as methionine and betaine. Clinical routine blood assays showed a mild, but significant increase in serum homocysteine levels in the NR group ($Δ = +1.66 ± 0.63\ \mu mol/L$; $p = 5.4 \times 10^{-4}$), but not in the placebo group ($Δ = −0.73 ± 1.7\ \mu mol/L$; $p = 0.869$, Supplementary Data 3), when comparing the last visit to baseline (Fig. 5a, b). A more detailed analysis including data from all study visits (V1-V7) indicated that serum HCy only increased at the second visit (V2, at day 3 of treatment; $Δ = +2.29 ± 1.72\ \mu mol/L$, paired t-test, $p = 0.002$), and subsequently remained stable until the end of the study ($Δ = −0.63 ± 1.76\ \mu mol/L$, paired t-test $p = 0.28$; Fig. 5c). For most participants, HCy levels remained within normal range. Three participants had slightly elevated HCy levels already at baseline, while one participant's HCy levels rose

from normal to slightly elevated levels. No participants developed any clinical symptoms related to the observed HCy increase. Moreover, serum levels of methylation-relevant metabolites such as folic acid, methylmalonic acid and vitamin B12 remained unchanged (Supplementary Data 3).

Targeted metabolomic analysis of whole blood did not show any HCy elevation (Fig. 5d), or any changes in the major methyl-group donor S-adenosyl-methionine (SAM), or its immediate demethylation product S-adenosyl-homocysteine (SAH; Fig. 5e–g).

Finally, untargeted metabolomic analysis showed that whole blood methionine levels (a precursor for SAM synthesis) were unaffected, while betaine, a substrate for betaine homocysteine methyltransferase catalyzing one way of methionine synthesis, decreased slightly in the NR group (Supplementary Data 6, Fig. 5j–k).

## Discussion

Our study met its primary outcome and established that orally administered NR at a dose of 3000 mg daily is well tolerated in PD over a course of 4 weeks. NR-recipients developed no moderate or severe adverse events. Furthermore, there were no mild adverse events likely attributed to NR, and no other clinical or biochemical signs of toxicity.

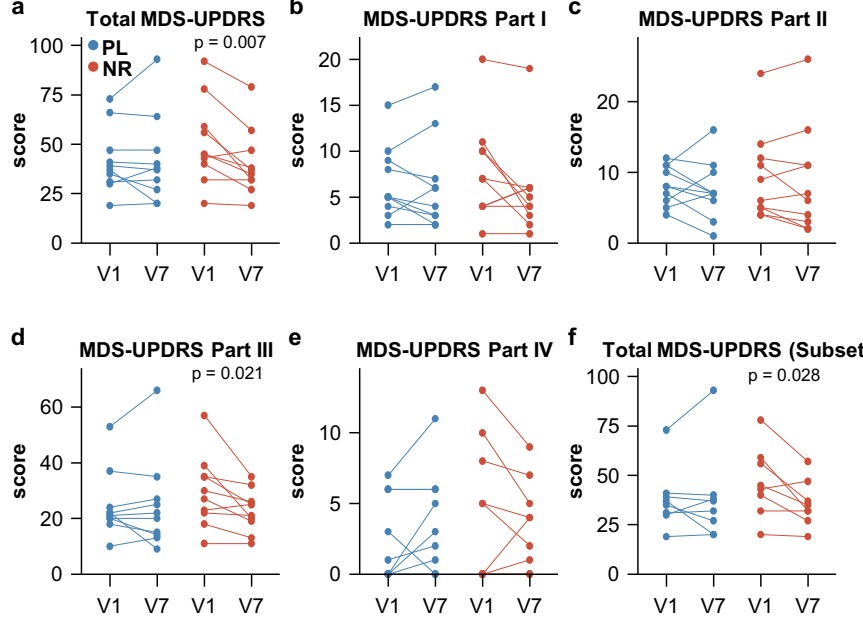

**Fig. 2 | High dose NR supplementation improves MDS-UPDRS scores in individuals with PD. a–e** Change in MDS-UPDRS scores for the placebo (blue) and NR (red) group, both for total MDS-UPDRS and parts I-IV of the MDS-UPDRS scale (NR group: *n* = 10, placebo group: *n* = 10). **f** Subset of patients (NR group: *n* = 8, placebo group: *n* = 8) with less different intervals since levodopa administration at the last visit. All *p*-values were generated using unadjusted, two-tailed paired t-tests. NR NR group, PL Placebo group, MDS-UPDRS Movement Disorder Society Unified Parkinson's Disease Rating Scale. Source data are provided as a Source Data file.

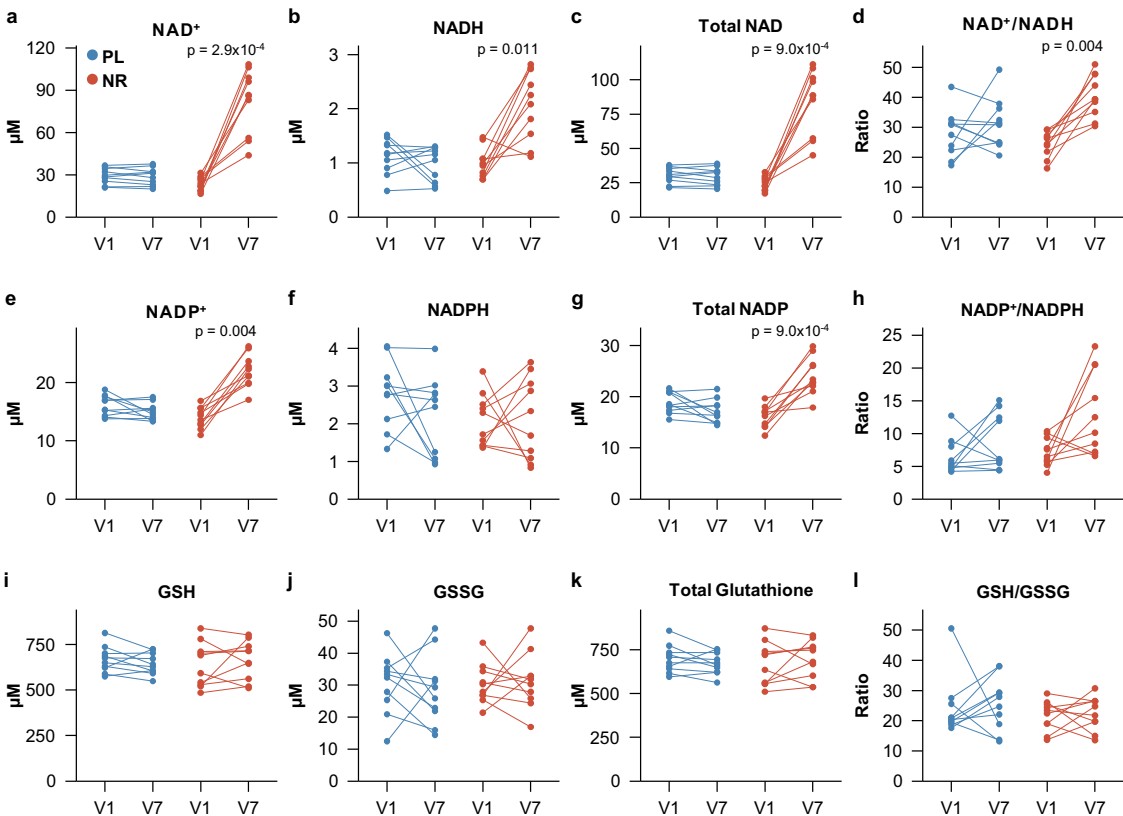

**Fig. 3 | High dose NR increases NAD and NADP metabolites in whole blood samples. a–l** Change from baseline (V1) to last visit (V7) in the placebo (blue) and NR (red) group for the indicated metabolites in flash frozen whole blood samples using the NADmed method (NR group: *n* = 10, placebo group: *n* = 10). All *p*-values were generated using paired two-tailed t-tests for normally distributed data and paired two-tailed Wilcoxon test for non-normally distributed data. All *p*-values were corrected using the Benjamini–Hochberg method. NAD+ Nicotinamide adenine dinucleotide (oxidized), NADP+ Nicotinamide adenine dinucleotide phosphate (oxidized), GSH Glutathione, GSSG Glutathione disulfide, NR NR group, PL Placebo group. Source data are provided as a Source Data file.

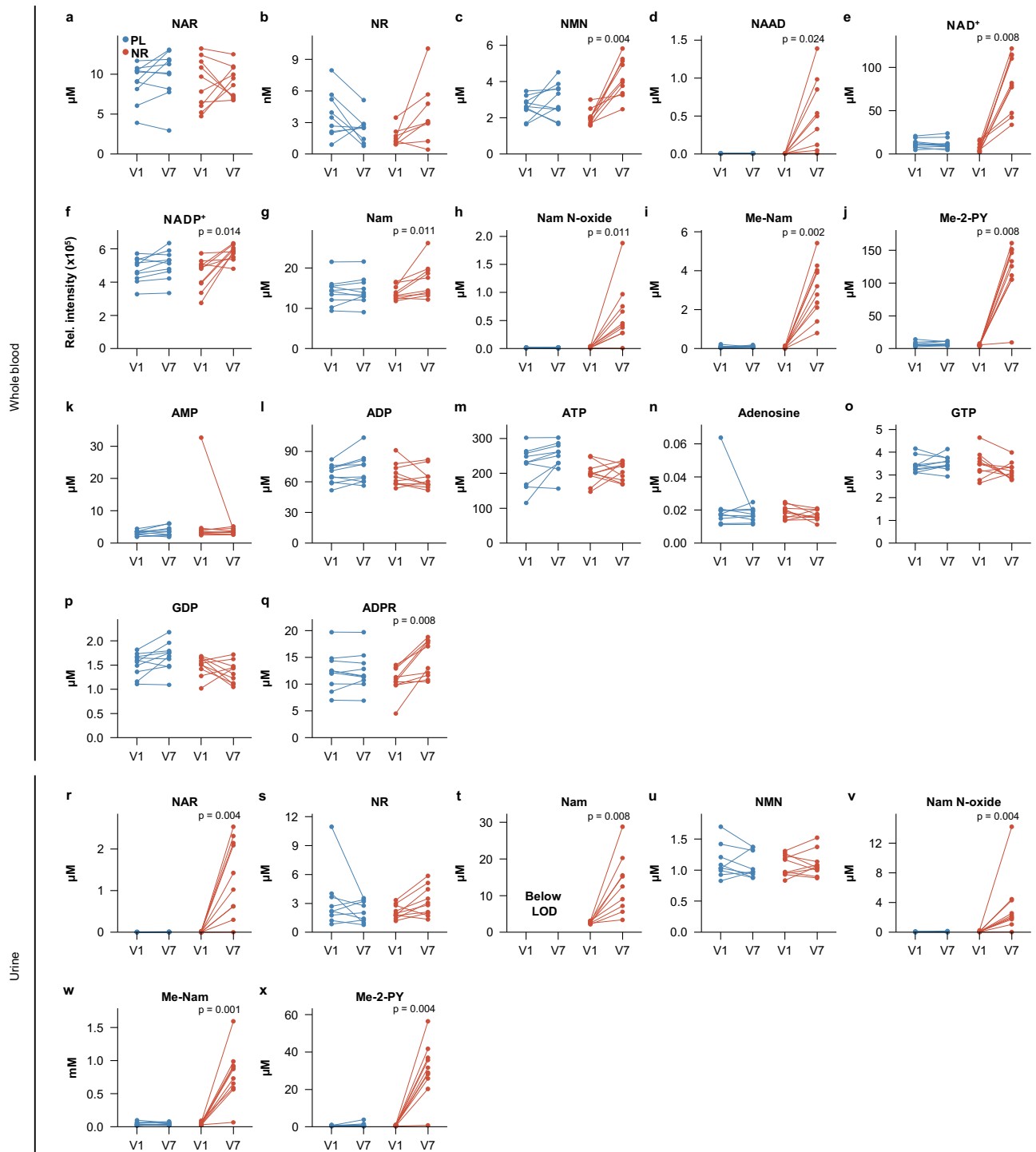

**Fig. 4 | High dose NR augments the NAD metabolome. a–q** Data from flash frozen whole blood samples analyzed by LC-MS for the indicated metabolites in the placebo (blue) and NR (red). NAAD: NR group: *n* = 9, placebo group: *n* = 10; NR NR group: *n* = 8, placebo group: *n* = 9; all other metabolites: NR group: *n* = 10, placebo group: *n* = 10. **r–x** Data from flash frozen urine samples analyzed by LC-MS, presented with the same color scheme as above. NAR: NR group: *n* = 10, placebo group: *n* = 8; Nam: NR group: *n* = 9, placebo group: all below limit of detection; all other metabolites: NR group: *n* = 10, placebo group: *n* = 9. All *p*-values were determined using paired two-tailed t-test for normally distributed data and paired two-tailed Wilcoxon test for non-normally distributed data. All *p*-values were corrected using

the Benjamini–Hochberg method. LOD Limit of detection, NAD⁺ Nicotinamide adenine dinucleotide (oxidized), NADP⁺ Nicotinamide adenine dinucleotide phosphate (oxidized), Me-Nam 1-methyl nicotinamide, NAAD nicotinic acid-adenine dinucleotide, Me-2-PY N1-methyl-2-pyridone-5-carboxamide, Nam Nicotinamide, Nam N-oxide Nicotinamide N-oxide, ADPR ADP-ribose, NAR Nicotinic acid riboside, NR Nicotinamide riboside, NMN Nicotinamide mononucleotide, NA Nicotinic acid, ATP Adenosine triphosphate, ADP Adenosine diphosphate, AMP Adenosine monophosphate, GDP Guanosine diphosphate, GTP Guanosine triphosphate, NR NR group, PL Placebo group. Source data are provided as a Source Data file.

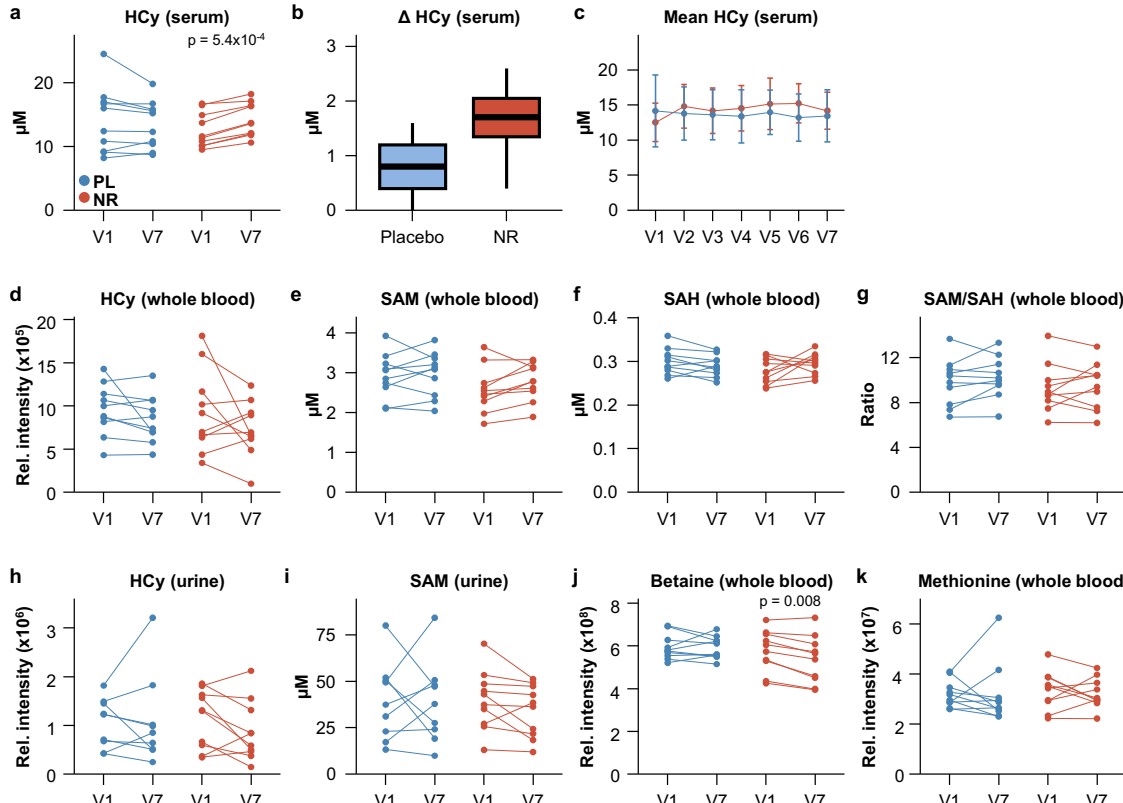

**Fig. 5 | High dose NR supplementation mildly increases homocysteine in serum, but not whole blood, and does not lead to methyl group pool depletion.**
**a** Change of serum HCy on the individual level from baseline (V1) to last visit (V7). NR group: $n = 10$, placebo group: $n = 10$. **b** Change of serum HCy on the group level from baseline (V1) to last visit (V7). Median (middle line), 25th and 75th percentiles (box), and 5th and 95th percentiles (whiskers). NR group: $n = 10$, placebo group: $n = 10$. $p$-values were determined using independent two-tailed t-test and corrected using the Benjamini–Hochberg method. **c** Mean serum HCy levels at all indicated visits. Error bars indicate standard deviation. NR group: $n = 10$, placebo group: $n = 10$. Reference serum values for HCy: 60 years of age: <15.9 μM; 41–60 years of

age: <14.2 μM. Level of HCy (**d**), SAM (**e**), SAH (**f**), and the SAM/SAH ratio (**g**) in whole blood samples at baseline (V1) and last visit (V7). NR group: $n = 10$, placebo group: $n = 10$. **h**, **i** Urine levels of HCy and SAM (NR group: $n = 10$, placebo group: $n = 9$). Change of betaine (**j**) and methionine (**k**) in whole blood samples (NR group: $n = 10$, placebo group: $n = 10$). Placebo group: blue; NR group: red. $P$-values shown in **a** and **c**–**k** were determined using paired two-tailed t-tests for normally distributed data and paired two-tailed Wilcoxon tests for non-normally distributed data. All $p$-values were corrected using the Benjamini–Hochberg method. HCy homocysteine, SAM S-adenosyl methionine, SAH S-adenosyl homocysteine, NR NR group, PL Placebo group. Source data are provided as a Source Data file.

While our data do not guarantee long-term safety, they allow future dose-optimization and efficacy studies to extend the tested dose range of NR to 3000 mg daily, provided that appropriate safety monitoring is implemented. In our opinion, adequate dose experimentation in clinical trials will be paramount, in order to adequately explore and exploit potential dose-dependent beneficial effects in PD, as well as the multitude of other neurological and non-neurological diseases, for which NR is currently being tested. Indicatively, there are 44 NR-related trials registered at www.clincaltrials.gov at the time of submission. Furthermore, our results suggest that NR doses up to 3000 mg daily do not need to be gradually increased and can be initiated directly in NR-naïve individuals.

NR-treatment improved clinical symptoms of PD, measured by MDS-UPDRS, while this remained unchanged in the placebo group. This improvement was driven mostly by MDS-UPDRS part III, which assesses motor function, and part I, which assesses non-motor aspects of daily living. Interestingly, a qualitatively similar but weaker effect was seen in the NADPARK trial where participants consumed 1000 mg NR daily[18]. Taken together, these observations suggest that NAD augmentation may have a symptomatic anti-Parkinson effect. However, since the treatment group also exhibited a shorter time from last dose of dopaminergic medication at the final visit compared to baseline, we cannot exclude that this may have influenced the observed change in the MDS-UDPRS. Nevertheless, it should be stressed that this part of the study was designed to detect potential worsening of the

parkinsonism, and not to assess symptomatic improvement. Thus, the finding of symptomatic improvement should be considered preliminary, awaiting confirmation in larger trials, such as the ongoing phase II trials N-DOSE[41] and NOPARK[42].

At a dose of 3000 mg daily, NR treatment greatly augmented the NAD-metabolome, including a ~3.7-fold (range 1.8–5.8-fold) increase in whole blood NAD+ levels. This increase is considerably higher compared to that reported with 1000 mg or 2000 mg NR daily, which increased blood NAD+ levels 1.37–2.8-fold[26,28,33,35,36]. However, in addition to the applied NR dose these studies also vary in terms of both sample type (e.g., whole blood, peripheral blood mononuclear cells or serum), and duration of exposure. Therefore, while providing an indication of a dose effect, the results of these studies are not fully comparable.

In addition to prominently elevated NAD+ levels, the NR-group exhibited a clear increase in the NAD+/NADH ratio, which has been reported to be decreased in PD[43]. Furthermore, NR supplementation was associated with an increase in whole blood NADP. An elevated NADP pool may be beneficial in several ways. NADP is a major redox factor in anabolic synthesis reactions, but also mediates cellular defense against oxidative stress, is involved in cytochrome P450 dependent detoxification mechanisms, and plays a role in immune response by generating oxidative bursts, among others[44,45]. With regard to PD, the reduced form of NADP, NADPH, has been shown to ameliorate MPTP-induced dopaminergic neurodegeneration in

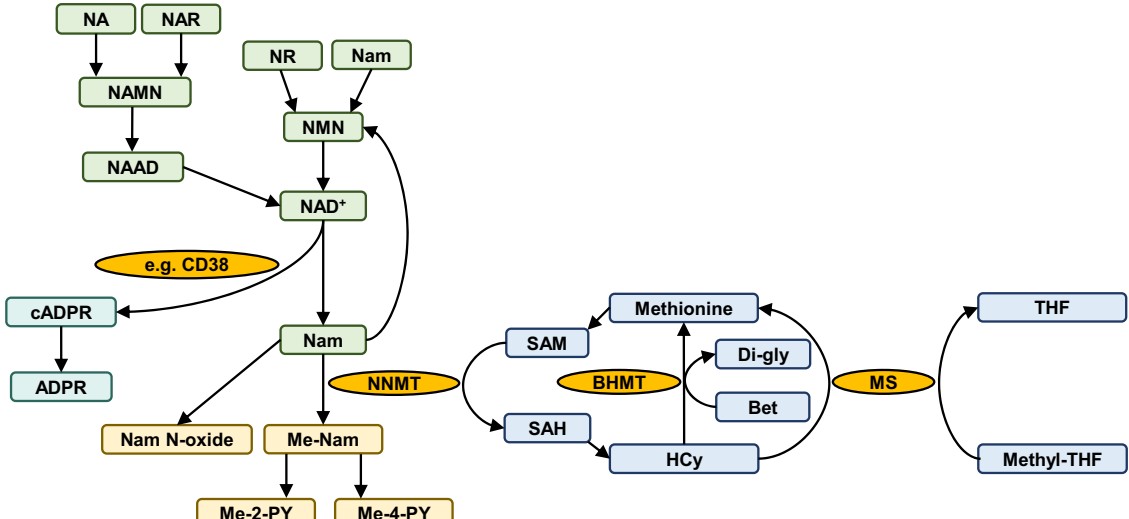

**Fig. 6 | Overview of NAD metabolism in interaction with methyl-group metabolism.** The figure shows various routes of NAD biosynthesis, and examples of NAD degradation leading to Nam production. If not recycled, Nam is methylated to Me-Nam, in a reaction catalyzed by NNMT and requiring SAM as methyl-group donor. The other product of this reaction, SAH, is further converted to HCy, and subsequently regenerated to methionine in the methionine cycle. Me-Nam 1-methyl nicotinamide, NAAD Nicotinic acid adenine dinucleotide, Nam Nicotinamide, Nam N-oxide Nicotinamide N-oxide, cADPR Cyclic ADP-ribose, ADPR Adenosine diphosphate ribose, NAR Nicotinic acid ribonucleoside, NR Nicotinamide riboside, NMN, Nicotinamide mononucleotide, NA Nicotinic acid, Me-2-PY N1-methyl-2-pyridone-5-carboxamide, Me-4-PY N-methyl-4-pyridone-5-carboxamide, NAMN Nicotinic acid mononucleotide, SAM S-adenosyl methionine, SAH S-adenosyl homocysteine, NNMT Nicotinamide N-methyltransferase, MS Methionine synthase, CD38 Cluster of differentiation 38/cyclic ADP-ribose hydrolase, Methyl-THF Methyl-tetrahydrofolate, THF Tetrahydrofolate, BHMT, Betaine homocysteine methyltransferase, Di-Gly Dimethyl glycine, Bet Betaine.

cultured cells[46], and to exert anti-neuroinflammatory and anti-neurotoxic effects in an MPTP-induced mouse model of PD[47]. Thus, it is possible that NADP augmentation may contribute to NR-associated neuroprotective effects in PD.

Finally, the observed increase in ADPR, a signaling molecule derived from NAD[+] by the action of CD38 and other enzymes also abundantly expressed in the brain[48,49], further supports a functional impact of NR-induced NAD augmentation on human metabolism.

NR treatment was associated with a mild but significant increase in serum HCy levels. A similar HCy increase was recently reported with escalating NR supplementation from 250 to 1000 mg over 5 months[33], but this was not seen following treatment with 1000 mg NR for one month. This suggests that the NR-associated HCy increase may be dose- and/or time-dependent. The observed increase in serum HCy levels in our data occurred quickly, during the first three days following commencement of treatment, and levels remained stable thereafter, suggesting that a new equilibrium had been reached. Interestingly, the increase in HCy was only observed in routine clinical serum biochemistry, and not reproduced in whole blood metabolomics by LC-MS. The reason for this discrepancy may be related to the sample material (i.e., serum vs whole blood), or to method sensitivity. Notably, the change observed in the serum, although statistically significant, was very mild. The observed increase in serum HCy is not unexpected. The NR-induced augmentation of NAD-metabolism leads to increased production of Nam. This is in turn converted to methyl-nicotinamide (Me-Nam) by the action of nicotinamide N-methyltransferase (NNMT), which transfers a methyl group from the universal methyl-donor S-adenosyl methionine (SAM) to Nam, also producing S-adenosylhomocysteine (SAH). SAH is then further converted to homocysteine as part of the homocysteine-methionine cycle (Fig. 6)[50]. Because of the dependence of Nam breakdown on methyl-group donation from SAM, it has been postulated that high dose NAD precursor supplementation may, in theory, lead to overconsumption of SAM and depletion of the methyl-group pool, which could result in an impairment of methylation homeostasis[40]. As we did not detect a change in HCy, SAM, SAH, the SAM/SAH ratio or methionine in whole blood, our data indicates that NR even at a dose of 3000 mg for

4 weeks does not cause depletion of the methyl-group pool. Slightly decreased levels of betaine, a substrate of one of two possible pathways for the re-conversion of HCy to methionine[51], were observed in the NR group, however, this did not affect methionine levels. It is possible that this betaine reduction indicates the higher flux for resynthesis of methionine and, eventually, SAM, due to the higher demand of methyl group for the methylation of Nam.

It has also been reported that NR treatment at lower doses, alone or in combination with pterostilbene, may be associated with a mild increase in triglycerides, total cholesterol and LDL[31,34]. We observed no such effects in our study. In fact, the NR group showed a nominal trend for a reduction in serum triglyceride levels. Furthermore, one previous study with NR in healthy middle-aged and older adults reported a trend for decreased systolic blood pressure[27]. While this was not corroborated by our findings, we note that PD patients typically have an impairment of the autonomic nervous system, including blood pressure regulation. Therefore, any blood pressure effects, or lack thereof, cannot be generalized on a population level.

Our study has several limitations. The results are based on a 4-week trial period and thus not informative with regard to long-term safety of 3000 mg NR daily. Therefore, until this is established, we strongly discourage the use of this dose outside of clinical studies with appropriate safety monitoring. Nevertheless, our results are sufficient to enable further testing of this dose in upcoming phase II clinical trials. The female to male ratio in our study, with only 3 and 2 females included per arm, was lower than that observed in the general PD population[52]. This makes our results less generalizable for women with PD. However, sex-specific effects have not been observed for NR. The observed symptom improvement in the NR group may be partly confounded by shorter time from last levodopa dose and must therefore be interpreted with care. However, the NR-SAFE trial was not designed to address clinical efficacy of NR in PD, this will be assessed by other ongoing phase II studies[41,42,53]. Since our study was conducted on individuals with PD, our findings are not necessarily generalizable to other diseases. Still, given the fundamental and ubiquitous nature of NAD-metabolism, we deem it unlikely that severe adverse events would emerge on a disease-specific basis. It is therefore unlikely that

separate safety trials would be required for other conditions, provided that adequate safety monitoring is included in the study.

## Methods

### Participants and study design

This study design was a phase I, single-center, randomized, double-blinded, placebo-controlled trial, aiming to provide initial evidence of safety and tolerability for NR at a dose of 3000 mg daily. The trial was conducted at the Department of Neurology, Haukeland University Hospital, Norway, from the 29.04.22 to the 01.07.22. The first participant was enrolled on 29.04.22 and the last on 27.05.2022. Inclusion criteria were: (i) age equal to or greater than 35 years and lower than 100 years at time of enrollment, (ii) clinical diagnosis of idiopathic PD according to the MDS criteria[54,55], and (iii) Hoehn and Yahr score <4 at time of enrollment[56]. Exclusion criteria were: (i) Dementia or other neurodegenerative disorders at baseline visit, (ii) any psychiatric disorder that would interfere with compliance in the study, (iii) any severe somatic illness that would make the individual unable to comply and participate in the study, (iv) use of vitamin B3 supplementation within 30 days prior to enrollment, and (v) metabolic, neoplastic or other physically or mentally debilitating disorders at baseline visit. Patient recruitment, inclusion, and follow-up was carried out by a GCP certified investigator. The research protocol was approved by the Regional Committee for Medical and Health Research Ethics, Western Norway (379218). Patients were identified and recruited at the Department of Neurology, Haukeland University Hospital, Norway. Written and informed consent was obtained from all participants from investigators in NR-SAFE. No financial or other compensation was offered to participants. This study was conducted according to Good Clinical Practice guidelines. The trial was registered on 25.04.2022 on www.Clinicaltrials.gov, identifier: NCT05344404. The study complies with ICMJE guidelines on reporting of clinical trials.

### Randomization and masking

20 participants (5 female and 15 male) were randomly allocated to receive either 1500 mg NR twice daily or placebo for 4 weeks (number of blocks: 2, block size 4, allocation: 1:1), using the electronic Case Report Form (www.viedoc.com, version 4.77.8648.13864). Sex was determined by assigned sex in the electronic patient records. NR was administered twice-daily to increase compliance, given the number of capsules participants were required to ingest, and to ensure the stability of the resulting increase in blood NAD levels. The drug containers were sequentially marked by an independent third party at the research and development department at Haukeland University Hospital. The study drug nicotinamide riboside chloride (NR; Tru Niagen) and placebo were provided by ChromaDex, USA. Each NR-capsule contained 250 mg of NR. Placebo capsules contained microcrystalline cellulose. The NR and placebo capsules were identical in flavor, color, smell and shape. Drug compliance was determined by self-reporting from participants at study visits and a pill count of remaining medication when providing new study medication and at the end of the study. The Center for Clinical Research, Haukeland University Hospital provided allocation sequence, packaging and labeling of drugs in participant-specific kits. Participants, examining physicians as well as medical- and research staff were blinded for the duration of the trial. The sample size was estimated at 10 participants in each group, based on the homogenous response to NR seen in our previous study[18].

### Primary outcome

The primary outcome was the safety of 3000 mg NR daily for 4 weeks, defined as the absence of clinically significant, NR-associated moderate or severe adverse events.

### Secondary outcomes

The secondary outcomes were the incidence of treatment-associated mild adverse events, treatment-associated changes in the NAD metabolome in blood and urine, and the change in the overall clinical severity of PD, measured by total MDS-UPDRS.

### Procedures

At screening for trial entry, candidates underwent physical and neurological examination. Enrolled participants were assessed by physical visits by a movement disorder specialist at baseline (visit 1, V1), day 7 (V4), day 14 (V5), day 21 (V6) and day 28 (V7). At baseline and day 28 participants underwent additional assessment with the Movement Disorder Society Unified Parkinson's disease Rating Scale section I-IV (MDS-UPDRS). At all physical visits an electrocardiogram (ECG), vital parameters (blood pressure, pulse, weight, height) and blood samples were obtained. Additional blood samples were taken on day 3 (V2) and day 5 (V3) to assess safety. Flash frozen whole blood samples for metabolite analyses (NADMed and LC-MS) were obtained by collecting blood in EDTA tubes, aliquoting into 250 μl aliquots and snap freezing the tubes in liquid nitrogen precisely 2 min after blood was drawn. The participants were interviewed by telephone consultation by a study nurse on day 3 and day 35 to screen for additional adverse events. Participants were instructed to take their anti-Parkinson drugs as they normally would and no changes were made to anti-Parkinson drug treatment during the trial. Adverse events (AEs) were recorded by the investigating physician. These were then classified by severity according to the Common Terminology Criteria for Adverse Events v5.0 (CTCAE)[57] as either mild (1), moderate (2), severe (3), life-threatening (4) or death (5). The AE relation to the study drug was scored as unrelated (1), unlikely (2), possible (3), probable (4) or definitely (5).

### Clinical laboratory values

Analysis of clinical laboratory values from blood samples was performed by the Department of Medical Biochemistry and Pharmacology (MBF) at Haukeland University Hospital, Bergen, Norway.

### NAD and glutathione metabolite analysis from frozen blood

Analysis was carried out by NADMed (Helsinki, Finland; www.nadmed.com). NAD and glutathione metabolites were extracted from frozen blood in a single step using a proprietary extraction procedure, and each metabolite was measured individually using optimized cyclic enzymatic assays with colorimetric detection. For stability, the NADMed extraction method utilizes an extraction solution representing a non-buffer complex mixture of organic solvents in water. Samples are injected into a pre-warmed extraction solution to force protein unfolding and release non-covalently bound metabolites into solution. All pH and redox sensitive metabolites remain stable, including reduced and oxidized glutathione. After protein removal, supernatants are used for separate measurements of total GSH and GSSG. GSH is determined as the difference of total GSH pool and GSSG values.

### LC-MS analysis from whole blood and urine

Sample analysis was carried out by MS-Omics (Vedbæk, Denmark).

Blood sample extraction was carried out by combining 100 μl whole blood with 900 μl ice-cold extraction solvent (Methanol/Acetonitrile/MQW – 10:6:4 fortified with stable isotope internal standards (Supplementary Data 6). After vortexing for 4 min and centrifugation for 10 min at $15,000 \times g$ and 4 °C, 900 μl of extract were added into a Phree filter, centrifuged for 10 min at $1800 \times g$ and 4 °C. 250 μl of the filtrate were transferred into two high-recovery LC-vials (one for semi-polar and one for polar analysis) and samples dried under a flow of nitrogen (-30 min).

Urine sample extraction was carried out by adding 50 μl of urine into a SpinX filter and mixing it with 50 μL eluent B (10 mM ammonium

acetate in 90 % acetonitrile). The filters were centrifuged at $3000 \times g$ and 4 °C for 10 min and the filtrate collected. Samples were subsequently diluted additionally 2.5 times using MQW and eluent B to achieve an acetonitrile content of 45% (40 μl sample extract, 30 μl MQW and 30 μl eluent B).

Samples were handled during the whole extraction procedure at 4 °C, and sample analysis was started immediately after sample preparation without any intermediate storage. To ensure high-quality sample preparation, a quality control sample (QC sample) was prepared by pooling small equal aliquots from each sample for all approaches, to create a representative average of the entire set. This sample was treated and analyzed at regular intervals throughout the sequence.

Polar metabolite profiling of whole blood and urine was performed using a Thermo Scientific Vanquish LC coupled to Thermo Q Exactive HF MS. An electrospray ionization interface was used as ionization source. Analysis was performed in negative and positive ionization mode. The UPLC was performed as described by Hsiao et al. [58], with the modification that deactivation additives were only added to the aqueous eluent and 5 μL were used as injection volume. Peak areas were extracted using Compound Discoverer 3.3 (Thermo Scientific).

Semi-polar metabolite analysis of whole blood was carried out using a Thermo Scientific Vanquish LC coupled to Orbitrap Exploris 240 MS, Thermo Fisher Scientific. An electrospray ionization interface was used as ionization source. Analysis was performed in positive and negative ionization mode under polarity switching. The UPLC was performed using a the protocol described by Doneanu et al. [59]. Peak areas were extracted using Compound Discoverer 3.3 (Thermo Scientific). The results from the two platforms were merged into one dataset. Compounds detected in both platforms were selected so the final data only contains the best representation of the compound.

For urine samples only polar extraction was performed. For blood samples the applied extraction method is indicated for each reported metabolite in Supplementary Data 6.

Identification of compounds for both methods was performed at four levels; Level 1: identification by retention times (compared against in-house authentic standards), accurate mass (with an accepted deviation of 3 ppm), and MS/MS spectra; Level 2a: identification by retention times (compared against in-house authentic standards), accurate mass (with an accepted deviation of 3 ppm). Level 2b: identification by accurate mass (with an accepted deviation of 3 ppm), and MS/MS spectra, Level 3: identification by accurate mass alone (with an accepted deviation of 3 ppm). All standard metabolites used here and the annotation level of the reported metabolites are indicated in Supplementary Data 6. For a few metabolites only relative intensities (areas) are provided. In the case of homocysteine and $NADP^+$, this is due to instability of the compound in the standard solutions. In the case of betaine and methionine, this is because those metabolites lacked a standard series for absolute quantification. Compounds with values available from less than 7 samples, were not included in downstream analyses.

### Statistical analysis

Data normality was tested by the Shapiro–Wilk test. The majority of variables in the dataset were normally distributed (Supplementary Data 7). Statistical comparison of continuous variables between baseline and end of study were conducted by two-tailed paired t-tests. Comparison of mean changes between the NR and placebo groups were performed by independent two-tailed t-tests. For values that were not normally distributed, paired and independent Wilcoxon tests were performed. Analysis of categorical variables was carried out with the Fisher's exact test. Correlations were calculated using Pearson's correlation. Multiple testing correction was performed with the Benjamini–Hochberg method. All statistical analyses were performed using R (https://cran.r-project.org, version 4.2.2, R Foundation for Statistical Computing, Vienna, Austria) using the following packages;

tidyverse version 1.3.1, readxl version 1.4.0, openxlsx version 4.2.5, lubridate version 1.8.0, cowplot version 1.1.1, ggpubr version 0.4.0, ggsignif version 0.6.3, rvg version 0.3.2, officer version 0.6.0 and effsize version 0.8.1.

### Reporting summary

Further information on research design is available in the Nature Portfolio Reporting Summary linked to this article.

## Data availability

The vital signs, MDS-UPDRS and NADMed analysis data required to reproduce the results presented in this manuscript have been deposited in the Neuromics Group repository https://git.app.uib.no/neuromics/nr-safe with no planned end date. Data can be accessed publicly by using the provided URL. Whole blood and urine metabolomics data have been deposited to the EMBL-EBI MetaboLights database[60] with the identifier MTBLS8812 (https://www.ebi.ac.uk/metabolights/MTBLS8812). The raw demographic, drug- and medical history data are protected and not available due to data privacy laws. Any additional information required to reanalyze the data reported in this paper is available from the corresponding authors upon request. Source data are provided with this paper.

## Code availability

The code required to reproduce the results presented in this manuscript have been deposited in the Neuromics Group repository https://git.app.uib.no/neuromics/nr-safe with no planned end date. Data can be accessed publicly by using the provided URL.

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

## Acknowledgements

We are deeply grateful to the patients involved in the study. We are grateful to study coordinator Ingunn Anundskås, and thank our technical laboratory staff, Martina Castelli, Sepideh Mostafavi, Yana Mikhaleva, Hanne Linda Nakkestad, and Gry Hilde Nilsen, for excellent technical assistance. We thank Prof Mathias Ziegler and Dr Brage Brakedal for the inspiring scientific discussions. We also thank Dr Ann Cathrine Kroksveen, Biobank Haukeland, Haukeland University Hospital. We are grateful to ChromaDex (Irvine, California) for providing the NR and placebo capsules for the study. The trial was supported by grants from The Research Council of Norway (288164, C.T.), Trond Mohn Foundation (BFS2017REK05, C.T.) and the Western Norway Regional Health Authority (F-11470 CT, F-12133 CT, F-12823 H.B.). The sponsor of the study is Haukeland University Hospital, who hosted the trial and provided the required facilities and personnel.

## Author contributions

H.B.: participated in the study conception and design, recruited and assessed study participants, collected, analyzed and interpreted data, and drafted the manuscript. S.K., G.O.S., K.H., E.S., M.S., S.A.G.: recruited and assessed study participants, and collected and interpreted data. C.D.: participated in study conception, design, data interpretation, data generation and performed critical revision of manuscript. C.T.: conceived, designed and directed the study, contributed to data analyses and interpretation, drafted the manuscript, and acquired funding for the study. All authors have read and approved the manuscript.

## Funding

## Competing interests

International patent applications relating to the use of nicotinamide riboside as a treatment for Parkinson's disease have been filed by the Technology Transfer Office "Vestlandets Innovasjonsselskap As (VIS)" on behalf of Haukeland University Hospital, Bergen, Norway, with publication authors C.T. and C.D. as inventors. The applications are pending and have the application numbers: PCT/EP2022/067408, PCT/EP2022/067412 and PCT/EP2023/060962. All other authors declare no competing interests.
