## [Peer Review File · Nature Communications]

REVIEWER COMMENTS

Reviewer #1 (Remarks to the Author):

This manuscript by Berven et al. reports the results of a double-blind, randomized placebo-controlled phase I clinical trial (NR-SAFE) which tested high dose (3,000 mg/day) nicotinamide riboside for 4 weeks in 20 people (n=10/group) with Parkinson's disease. The primary outcome was safety, assessed as the incidence of moderate and severe treatment-emergent adverse events (AEs). Secondary outcomes included incidence of mild AEs, changes in the blood and urine NAD⁺ metabolome, and change in clinical severity of the disease (assessed by the MDS-UPDRS). The study was rigorously conducted and adds to the existing literature surrounding overall tolerability of NR, including the addition of new data at a higher dose. Primary concerns are related to the small sample size and short duration, which is likely too small and short for detecting group differences in AEs and could lead to over-interpretation of the clinical relevance of the data.

Specific Comments:

Introduction; Line 69: It would be helpful to provide some brief context as to why NAD⁺ levels decline in blood with age and how raising NAD⁺ is specifically thought to be beneficial for PD (i.e., what are the underlying mechanisms through which it may act to improve outcomes in PD).

Results; Line 151: The MDS-UPDRS outcomes provide interesting preliminary evidence of efficacy but do not appear to have been corrected for multiple comparisons. Effect sizes should be reported to confirm that the authors are sufficiently powered to test their hypotheses regarding clinical severity, as this was a secondary outcome of the trial and there does not appear to have been an a priori power calculation to support the sample size. Relatedly, the statement in the Discussion; Line 244 should be softened regarding the "significant improvement" of clinical symptoms to convey the fact that this was a secondary outcome and needs to be confirmed in Phase II studies with a greater focus on clinical severity as the primary end point.

Results; Line 164: The text indicates no difference in time from last levodopa dose in a subgroup of N=8 participants, yet there is a mean difference of 22 minutes between the NR and placebo groups included in this subset. This is either a typographical error in the reporting of the time interval in one of the groups (e.g., misplaced decimal), or the lack of significance between groups is driven by the very large standard deviation among participants. If the reported data are correct, then the subgroups do not appear to be well matched and the authors cannot rule out any temporal effect of levodopa on the study outcomes. If this is an error (e.g., decimal point in the wrong location) then it should be corrected.

Results; Line 170: It might make more sense to describe the NAD metabolome data after the primary safety results and before the evidence of clinical benefit, to better match the order in which these outcomes are presented in the introduction.

Results; Line 216: Recommend either deleting the word “only” or deleting the word “already” from this sentence as it appears redundant.

Discussion; Line 314: Reference #21 reported a decrease in systolic blood pressure (not an increase) in healthy middle-aged and older adults. This should be corrected. The effect observed in reference #21 was more pronounced in participants with elevated baseline blood pressure which may account for differences with the present study as the NR group was largely normotensive.

Methods; Line 331: It would be helpful for the reader to also list the clinicaltrials.gov identification number in the first paragraph of the methods.

Methods; Line 344: The clinicaltrials.gov registration for this study indicates that the study drug was split into two daily doses of 1500mg each. This should be acknowledged in the methods and in the abstract. A justification for splitting the dose should also be provided in the methods.

Methods; Line 345: The authors describe a blocked randomization scheme with 4 blocks but do not provide detail on what factors were used to determine the blocks.

Methods; Line 355: The information provided for sample size estimation is insufficient for determining whether the study was adequately powered to assess group differences in moderate and severe AEs. It will likely take a large number of participants (i.e., hundreds) before moderate and severe AEs are reported at a high enough frequency to discern group differences. At best, this study provides initial evidence of tolerability of 3,000 mg/day in people with PD and potentially justifies advancement into a Phase II trial.

Methods; Line 361: The secondary outcomes listed here differ from elsewhere in the paper and do not mention clinical severity or homocysteine.

Methods; Line 416: In general, it would be helpful to describe when correction for multiple testing was or was not used as not all of the data in the supplemental tables appear to have this correction.

Supplemental Table S3 could be referenced in the first section of the results as, other than the change in homocysteine, there were no abnormal changes in blood chemistries associated with high dose NR, which provides further evidence of safety.

Reviewer #2 (Remarks to the Author):

Nicotinamide adenine dinucleotide (NAD) replenishment therapy with nicotinamide riboside (NR) has shown promise results on Parkinson’s disease (PD). Yet, the safety range and optimal dose of NR therapy for PD remain undetermined, and doses higher than 2000 mg daily have not been tested on humans.

Berven and colleagues assess the safety of high-dose nicotinamide riboside (NR) therapy in a double-blind, randomized phase I trial of 3000 mg NR daily in PD. The primary safety outcome of the study was the incidence of treatment-associated moderate and severe adverse events (AEs). These adverse events were assessed by investigators and stated in the protocol (Assessment of Intensity, with 3 grades, 2/3 are registered AEs). No adverse events or signs of toxicity that were likely attributed to NR were observed, which could inform potential dose-dependent beneficial effects of NR in PD and other disorders in future dosing trials. Results suggest that NR doses of 3000 mg daily could be the starting dose for these studies. Well conducted study and interesting data with appropriate interpretations. Please see below my comments.

- Protocol of the study was submitted but statistical analysis plan SAP was not provided. It is good practice to submit SAP even for phase 1 studies.
- Given blood samples were collected for liver, kidneys, testes etc, good to provide these blood chemistry data in the manuscript.
- All stats seem sound
- Methods and materials could be placed before results.
- No need for p-values in table 1 as descriptive

Reviewer #3 (Remarks to the Author):

Berven et al, Nat Comm 2023

Berven and colleagues submitted a report on an intervention study with high dose of nicotinamide riboside chloride to Parkinson's disease patients.

Whilst the study is certainly interesting and it seems to show potential benefits – and positive safety outcomes – in this population, its report on the methodologies used is highly incomplete.

1. numbering the pages of the manuscript would have made reviewing easier.
2. given the interest of NAD+ boosters in different health conditions and pathologies, this study assumes many terms and parameters specific to the Parkinson's Disease (PD) community, that are not obvious to other scientific fields. For example, in the abstract: what is MDS-UPDRS? It is worth mentioning that levodopa is a commonly used drug in the treatment of PD. And throughout the text other acronyms are assumed to be common knowledge, that for the generalist readership of Nature Communications probably should not be assumed as so, e.g. What is HCG? what is the Hoehn and Yahr parameter and how to interpret it, etc.
3. This clinical study remains a small study, so some impression of power calculation would have been really beneficial to report on. How robust and translatable are these results in other PD populations...?
4. Perhaps the area of most concern are the Materials and Methods of the performed chemical analysis, These are highly incomplete: It is mentioned that 2 LC-MS methods were used based on refs 49 and 50, but the description of these methods is lacking. This is of high relevance for the manuscript, as many metabolites from the NAD+ metabolome are reported in the results. It is important to describe the methods used (chromatography conditions, MS conditions) and modifications applied to the published 2

methods. It is also important to mention which metabolites were reported in Figure 4 and by which method, which metabolites overlapped between methods, and if so, why one value was reported (and not the other), which identification level used per which metabolite. Sample preparation is completely missing and should be described. Which internal standards were used? Were the samples normalized for total protein amount? How did the authors deal with natural dilution effect of urine per subject – was normalization to creatinine applied or other...? Quantification procedures should be described. What were the validation procedures applied per method in order to be confident on the metabolite quantification? Which standard compounds were used and where were they purchased, etc etc.... And how was the stability of samples for NAD+ metabolome analysis handled? What were the pre-analytical conditions?

For GSH and SAM cycle measurements the same lack of reporting on methodologies... Here the concern on preserving reduced an oxidized glutathione may be even more of a concern – how did the authors proceeded in these measurements? Normally for clinical applications, oxidation of glutathione is conducted, depleting the reduced pool and leading to the report of total glutathione instead...

5. In the results in Fig 3 – what is NAD pool? In Fig 4 – how did the authors identity Me2PY? how certain that it is not Me-4 PY or a combination of these 2? Some metabolites have absolute quantification values expression in micro- / nano-molar, and other as relative intensity. This is not explained or justified....Validation of analytical methods when reporting quantitative values is essential, was not mentioned in this study.

6. In the discussion, the relationship between methyl pool (through measurement of SAM cycle intermediates) and NAD+ metabolome is not explored, even if these measurements were acquired...

7. This is obviously a great deal of inter-individual variability in a clinical study. It is important to reflect and discuss this in the Discussion and Limitation sections. Is this high dose of NR really going to improve PD patients at a more general perspective beyond 20 subjects?

Point-by-point reply to the Reviewers' Comments:

Reviewer #1

This manuscript by Berven et al. reports the results of a double-blind, randomized placebo-controlled phase I clinical trial (NR-SAFE) which tested high dose (3,000 mg/day) nicotinamide riboside for 4 weeks in 20 people (n=10/group) with Parkinson's disease. The primary outcome was safety, assessed as the incidence of moderate and severe treatment-emergent adverse events (AEs). Secondary outcomes included incidence of mild AEs, changes in the blood and urine NAD⁺ metabolome, and change in clinical severity of the disease (assessed by the MDS-UPDRS). The study was rigorously conducted and adds to the existing literature surrounding overall tolerability of NR, including the addition of new data at a higher dose. Primary concerns are related to the small sample size and short duration, which is likely too small and short for detecting group differences in AEs and could lead to over-interpretation of the clinical relevance of the data.

We thank the reviewer for the positive and thorough evaluation of our work and for the insightful comments and are addressing these point-by-point below. The reviewer's concerns regarding the study sample size and duration are addressed in our responses to the individual comments below.

Comment-1:

Introduction; Line 69: It would be helpful to provide some brief context as to why NAD⁺ levels decline in blood with age and how raising NAD⁺ is specifically thought to be beneficial for PD (i.e., what are the underlying mechanisms through which it may act to improve outcomes in PD).

Response:

We thank the reviewer for this comment and would be happy to add more background on this matter. The mechanisms underlying the age-related decline of NAD are not fully understood. However, a number of processes have been proposed to contribute, including increased expression of CD38, an NAD glycohydrolase which degrades NAD, decreased expression of NamPT, catalyzing the rate limiting step of NAD biosynthesis from nicotinamide via the salvage pathway, and increased DNA damage and repair leading to a higher NAD consumption rate via the action of Poly(ADP-ribose) polymerase 1 (PARP1).

Due to space limitations and the fact that these are but few of the proposed mechanisms potentially contributing to the age-dependent NAD decline, we would respectfully prefer referring to a number of excellent reviews on the topic, rather than considerably expanding this section of the introduction. We are, however, happy to add a section to the manuscript if deemed essential. Regarding the potential beneficial effects of NAD-replenishment therapy for PD, we have added a short summary accompanied by appropriate sources (please see lines 82-91)

Comment-2:

Results; Line 151: The MDS-UPDRS outcomes provide interesting preliminary evidence of efficacy but do not appear to have been corrected for multiple comparisons. Effect sizes should be reported to confirm that the authors are sufficiently powered to test their hypotheses regarding clinical severity, as this was a secondary outcome of the trial and there does not appear to have been an a priori power calculation to support the sample size. Relatedly, the statement in the Discussion; Line 244 should be softened regarding the “significant improvement” of clinical symptoms to convey the fact that this was a secondary outcome and needs to be confirmed in Phase II studies with a greater focus on clinical severity as the primary end point.

Response:

We apologize for the confusion regarding multiple testing correction. We did not perform multiple testing correction, because this secondary outcome was predefined as the change in the total MDS-UPDRS (i.e., not in the subparts of MDS-UPDRS). Thus, the primary statistical test was performed to test this predefined specific question, as described in the study protocol (and on clinicaltrials.gov). Having detected a significant difference in the total MDS-UPDRS, we then proceeded with a posthoc analysis to assess the subparts of MDS-UPDRS, and determined which of them were driving the observed change. Since the subparts of MDS-UPDRS are highly correlated with each other, we believe that the probability for type I error is very small in this case, whereas performing multiple testing correction would substantially increase the chance of type II error. Nevertheless, addressing the reviewer’s request, we have now performed multiple testing correction and included this in the paper, Tables, and the supplementary material.

Furthermore, we realize that the predefined MDS-UPDRS outcome and testing strategy were not adequately explained in the manuscript and have amended this by elaborating and clarifying under the Results (please see lines 185-87), as well as in the Section “Secondary outcomes” in the Methods (please see lines 416-421). We hope that these changes have provided the required clarity.

Effect sizes were not calculated, because power estimation is typically performed for the primary outcome of a trial. This being a phase I safety trial, the primary outcome was **descriptive**, i.e., we sought to determine whether 3000mg NR daily induces clinically relevant AEs in the study period, in order to establish whether it is clinically acceptable to employ this dose in further phase II trials. In other words, the question asked is whether NR induced **any** clinically significant severe or moderate adverse events – not the statistical difference in the incidence of adverse events between the groups (that can only be answered in a phase II-III trial). For this reason, there are no statistical analyses of the primary outcome. This setting is typical for phase I safety trials and is clearly stated in our predetermined outcomes in the protocol and on clinicaltrials.gov.

The sample size employed in the NR-SAFE trial is typical for this type of study. The trial was not designed to determine the long-term safety of 3000mg NR daily. This would require large sample sizes and longer exposure time in phase II, and then phase III and IV studies with hundreds to thousands of participants. Based on the results of the NR-SAFE study it will now be possible to proceed with a phase II trial of 3000mg NR daily.

The MDS-UPDRS was included as a secondary outcome to detect any clinically significant worsening of PD-symptoms (i.e., not necessarily to detect an improvement). Nevertheless, a statistically significant improvement is detected with a type I error probability of 0.7%, which indicates that the observed MDS-UPDRS change in the NR-group is highly unlikely to be by chance. The effect size for this comparison is relatively high (Cohen's $d = 1.095$). Given the design of our trial (as explained above) we have chosen not to include effect sizes for all statistical tests performed, as this would be very unusual for phase I safety trials and could be somewhat misleading.

Regarding the statement in the discussion, this has been toned down, as per the reviewer's request (please see lines 284-301). Additionally, a comment about further phase II study has been added to the limitations section (please see line 366). The discussion clearly states that this was a phase I trial (please see lines 297-301).

Comment-3:

Results; Line 164: The text indicates no difference in time from last levodopa dose in a subgroup of $N=8$ participants, yet there is a mean difference of 22 minutes between the NR and placebo groups included in this subset. This is either a typographical error in the reporting of the time interval in one of the groups (e.g., misplaced decimal), or the lack of significance between groups is driven by the very large standard deviation among participants. If the reported data are correct, then the subgroups do not appear to be well matched and the authors cannot rule out any temporal effect of levodopa on the study outcomes. If this is an error (e.g., decimal point in the wrong location) then it should be corrected.

Response:

We thank the reviewer for this observation. We fully agree that, given the small sample sizes and large interindividual variation, the possibility that levodopa contributes to the observed effect cannot be confidently excluded. We have amended the text under Results and Discussion and toned down the interpretation of this observation (please see lines 193-208 and 289-301).

Comment-4:

Results; Line 170: It might make more sense to describe the NAD metabolome data after the primary safety results and before the evidence of clinical benefit, to better match the order in which these outcomes are presented in the introduction.

Response:

We thank the reviewer for this suggestion. We have changed the order in the introduction to better reflect the order in the results.

Comment-5:

Results; Line 216: Recommend either deleting the word “only” or deleting the word “already” from this sentence as it appears redundant.

Response:

We thank the reviewer for this suggestion. We have removed the word “already” in the manuscript.

Comment-6:

Discussion; Line 314: Reference #21 reported a decrease in systolic blood pressure (not an increase) in healthy middle-aged and older adults. This should be corrected. The effect observed in reference #21 was more pronounced in participants with elevated baseline blood pressure which may account for differences with the present study as the NR group was largely normotensive.

Response:

We thank the reviewer for pointing out this error and have corrected it (line 356). Regarding comparing our findings to those of the referenced study, one important difference is that PD patients typically have an impairment of the autonomic nervous system, including blood pressure regulation. Any blood pressure effects, or lack of, in our trial can therefore not be generalized to the population. We have added this information to the Discussion (please see lines 357-360).

Comment-7:

Methods; Line 331: It would be helpful for the reader to also list the clinicaltrials.gov identification number in the first paragraph of the methods.

Response:

This has been amended in the manuscript in the first paragraph of the methods section (please see lines 392-393).

Comment-8:

Methods; Line 344: The clinicaltrials.gov registration for this study indicates that the study drug was split into two daily doses of 1500mg each. This should be acknowledged in the methods and in the abstract. A justification for splitting the dose should also be provided in the methods.

Response:

NR was administered twice-daily to increase compliance given the large number of capsules participants were required to ingest, and to increase the stability of blood NAD levels. This has been added to the abstract and methods sections (please see lines 396-400)

Comment-9:

Methods; Line 345: The authors describe a blocked randomization scheme with 4 blocks but do not provide detail on what factors were used to determine the blocks.

Response:

We thank the reviewer for this comment. The randomization scheme did not consist of 4 blocks, but 2 blocks, each of size 4. We have specified this in the methods section for further clarity (please see line 397).

Comment-10:

Methods; Line 355: The information provided for sample size estimation is insufficient for determining whether the study was adequately powered to assess group differences in moderate and severe AEs. It will likely take a large number of participants (i.e., hundreds) before moderate and severe AEs are reported at a high enough frequency to discern group differences. At best, this study provides initial evidence of tolerability of 3,000 mg/day in people with PD and potentially justifies advancement into a Phase II trial.

Response:

As stated in the paper, the NR-SAFE trial was conducted to assess initial safety and tolerability evidence for NR at a dose of 3000mg per day – not to detect group differences in moderate and severe AEs (which is why no formal power estimation was performed for the study). Thus, the sample size for this trial was empirically chosen and is rather typical for a phase I safety trial (most of which employ < 20 subjects).

Being a phase I safety trial, the primary outcome was **qualitative/descriptive**, i.e., we sought to determine whether 3000mg NR daily induces clinically relevant AEs in the study period, in order to establish whether it is clinically acceptable to employ this dose in further phase II trials, and which, toxicity parameters should be monitored in that case. The primary objective was, therefore, to identify whether NR induced **any** clinically significant severe or moderate adverse events in a 4-week period – not to assess statistical differences in the incidence of adverse events between the groups. This setting is typical for phase I safety trials and is clearly stated in our predetermined outcomes in the protocol and on clinicaltrials.gov. Based on the results of the NR-SAFE, it will now be possible to proceed with a phase II trial of 3000mg NR daily.

We agree with the reviewer that the clarity of these premises could be improved in the manuscript. We have added the sentence “This study design was a phase I, single-center, randomized, double-blinded, placebo-controlled trial, aiming to provide initial evidence of safety and tolerability for NR at a dose of 3,000 mg daily” to the methods “participants and study design” section (please see lines 377-378).

Comment-11:

Methods; Line 361: The secondary outcomes listed here differ from elsewhere in the paper and do not mention clinical severity or homocysteine.

Response:

We thank the reviewer for pointing this out and apologize for the confusion. We have corrected the relevant text.

Comment-12:

Methods; Line 416: In general, it would be helpful to describe when correction for multiple testing was or was not used as not all of the data in the supplemental tables appear to have this correction.

Response:

The following line has been added to the relevant supplementary tables where relevant: “P-adj denotes multiple testing correction via the Benjamini-Hochberg method”. For tables only listing unadjusted p-values, the following line has been added: “P-values were not corrected for multiple testing”.

Comment-13:

Supplemental Table S3 could be referenced in the first section of the results as, other than the change in homocysteine, there were no abnormal changes in blood chemistries associated with high dose NR, which provides further evidence of safety.

Response:

A reference to Supplementary Data 3 has been added (please see lines 176-178). We kindly thank the reviewer for the positive comments and thorough review.

Reviewer #2

Nicotinamide adenine dinucleotide (NAD) replenishment therapy with nicotinamide riboside (NR) has shown promise results on Parkinson's disease (PD). Yet, the safety range and optimal dose of NR therapy for PD remain undetermined, and doses higher than 2000 mg daily have not been tested on humans.

Berven and colleagues assess the safety of high-dose nicotinamide riboside (NR) therapy in a double-blind, randomized phase I trial of 3000 mg NR daily in PD. The primary safety outcome of the study was the incidence of treatment-associated moderate and severe adverse events (AEs). These advert events were assessed by investigators and stated in the protocol (Assessment of Intensity, with 3 grades, 2/3 are registered AEs). No adverse events or signs of toxicity that were likely attributed to NR were observed, which could inform potential dose-dependent beneficial effects of NR in PD and other disorders in future dosing trials. Results suggest that NR doses of 3000 mg daily could be the starting dose for these studies.

Well conducted study and interesting data with appropriate interpretations. Please see below my comments.

We thank the reviewer for the insightful comments and are addressing these point-by-point below.

Comment-1:

Protocol of the study was submitted but statistical analysis plan SAP was not provided. It is good practice to submit SAP even for phase 1 studies.

Response:

The reason a statistical analysis plan was not provided is that, this being a phase I safety and tolerability trial, the primary outcome was addressed in a qualitative, descriptive approach. There was, therefore, no power calculation or statistical testing for the primary outcome.

The NR-SAFE trial was not conducted to assess group differences in moderate and severe AEs, but to assess initial safety and tolerability evidence. Thus, the sample size for this trial was empirically chosen and is rather typical for a phase I safety trial. The primary outcome was descriptive, i.e., we sought to determine whether 3000 mg NR daily induces any clinically relevant AEs in the study period, in order to establish whether it is clinically acceptable to employ this dose in further phase II trials. Based on our results, it will now be possible to proceed with a phase II trial of 3000mg NR daily.

To improve clarity, we have added the sentence "This study design was a phase I, single-center, randomized, double-blinded, placebo-controlled trial, aiming to provide initial evidence of safety and tolerability for NR at a dose of 3,000 mg daily" to the methods "participants and study design" section (lines 377-378).

Nevertheless, we acknowledge the reviewer's point that an SAP could have been provided for the secondary outcomes. Unfortunately, this was not done.

Comment-2:

Given blood samples were collected for liver, kidneys, testes etc, good to provide these blood chemistry data in the manuscript.

Response:

All of these values have been included in Supplementary Data 3. Due to space limitations, we did not include a table in the manuscript.

Comment-3:

All stats seem sound

Response:

We thank the reviewer for this evaluation.

Comment 4:

Methods and materials could be placed before results.

Response:

We thank the reviewer for the suggestion. The order of the sections is, however, dictated by the journal style.

Comment 5:

No need for p-values in table 1 as descriptive

Response:

The reason we included p-values was to highlight that there were no differences between demographic parameters at baseline. We thank the reviewer for the positive comments.

Reviewer #3

Berven and colleagues submitted a report on an intervention study with high dose of nicotinamide riboside chloride to Parkinson's disease patients. Whilst the study is certainly interesting and it seems to show potential benefits – and positive safety outcomes – in this population, its rt on the methodologies used is highly incomplete.

Comment 1:

1. numbering the pages of the manuscript would have made reviewing easier.

Response:

Page numbering has been added to the manuscript.

Comment 2:

2. given the interest of NAD+ boosters in different health conditions and pathologies, this study assumes many terms and parameters specific to the Parkinson's Disease (PD) community, that are not obvious to other scientific fields. For example, in the abstract: what is MDS-UPDRS? It is worth mentioning that levodopa is a commonly used drug in the treatment of PD. And throughout the text other acronyms are assumed to be common knowledge, that for the generalist readership of Nature Communications probably should not be assumed as so, e.g. What is HCG? what is the Hoehn and Yahr parameter and how to interpret it, etc.

Response:

All abbreviations have been spelled out the first time they are mentioned in the manuscript. Furthermore, we have elaborated on the nature and usage of certain terms and items, such as levodopa, Hoehn and Yahr scale, what the relevant parts of the MDS-UPDRS measure, etc. We hope this has improved the clarity of the manuscript and rendered these terms less esoteric.

Comment 3:

3. This clinical study remains a small study, so some impression of power calculation would have been really beneficial to report on. How robust and translatable are these results in other PD populations...?

Response:

The NR-SAFE trial was not conducted to statistically assess between-group differences in the incidence of adverse events, but to assess initial safety and tolerability of 3000 mg NR per day. The primary outcome was descriptive, i.e., we sought to determine whether 3000 mg NR daily induces **any** clinically relevant AEs in the study period, in order to establish whether it is clinically acceptable to employ this dose in further phase II trials. Thus, a power estimation is not feasible in this case. This setting is typical for phase I safety trials. Based on our results, it will now be possible to proceed with a phase II trial of 3000mg NR daily. The population sizes employed in phase II, and

potentially further phase III and IV trials (i.e., typically well over 1,000) will enable such statistics to be conducted and be informative on the incidence adverse events, generalizable to the general PD-population.

To improved clarity, we have added the sentence “This study design was a phase I, single-center, randomized, double-blinded, placebo-controlled trial, aiming to provide initial evidence of safety and tolerability for NR at a dose of 3,000 mg daily” to the methods “participants and study design” section (please see lines 377-378).

Comment 4:

4. Perhaps the area of most concern are the Materials and Methods of the performed chemical analysis, These are highly incomplete: It is mentioned that 2 LC-MS methods were used based on refs 49 and 50, but the description of these methods is lacking. This is of high relevance for the manuscript, as many metabolites from the NAD⁺ metabolome are reported in the results. It is important to describe the methods used (chromatography conditions, MS conditions) and modifications applied to the published 2 methods. It is also important to mention which metabolites were reported in Figure 4 and by which method, which metabolites overlapped between methods, and if so, why one value was reported (and not the other), which identification level used per which metabolite. Sample preparation is completely missing and should be described. Which internal standards were used? Were the samples normalized for total protein amount ? How did the authors deal with natural dilution effect of urine per subject – was normalization to creatinine applied or other...? Quantification procedures should be described. What were the validation procedures applied per method in order to be confident on the metabolite quantification? Which standard compounds were used and where were they purchased, etc etc.... And how was the stability of samples for NAD⁺ metabolome analysis handled? What were the pre-analytical conditions?

For GSH and Sam cycle measurements the same lack of reporting on methodologies... Here the concern on preserving reduced an oxidized glutathione may be even more of a concern – how did the authors proceeded in these measurements? Normally for clinical applications, oxidation of glutathione is conducted, depleting the reduced pool and leading to the report of total glutathione instead...

Response:

a) We have now adjusted the method description and included changes where modifications were made to the referenced methods. For the polar extraction, this entailed that deactivation additives were only added to the aqueous eluent and 5µL were used as injection volume (please see line 474). For the semipolar method, the modifications did not concern the method itself, but the sample preparation. This has been corrected and the applied sample preparation has been added to the manuscript (see below).

b) The information about the annotation level of each reported metabolite has been added to Supplementary Data 6 and is referred to in the methods section (please see line 493). The information about which extraction method has been applied for each reported metabolite from

whole blood is now indicated in Supplementary Data 6 and referred to in the methods section (lines 485-486). Urine samples were only analyzed using the polar extraction method, which also has been clarified in the methods section (line 485).

c) Information about sample preparation has now been added to the method description (please see lines 445-451 and 454-469).

d) The internal standards were used for QC check of retention times, accurate mass, and intensity, to validate that extraction, dilution and instrumental performance were performing as expected. The following internal standards were used. Carnitine-d9 (Sigma, Cat#93689), Cholic acid-d4 (Cambridge Isotope Laboratories, Cat# DLM-2611-0), Glutamic acid-d5 (Sigma, Cat# 616281), Leucine-d10 (Sigma, Cat#492426), and Phenylalanine-d8 (Sigma, Cat# 490148). This information has been added to the methods section as well as to Supplementary Data 6 in the new sheet "Standard compounds".

e) Blood samples were not normalized to protein amount. Instead, the same volume was used for extraction from each sample and values expressed in molarity of the original sample. Similarly, for urine samples, potential dilution levels were not accounted for. Comparison was carried out for each participant individually before and after treatment, and samples were obtained in similar fashion at each timepoint.

f) Quantifications were performed using regression models based on external calibration rows.

g) With regard to the validation procedures applied to LC-MS analysis, please see the third answer to point 5 below.

h) The standard compounds and supplier have now been added to the new Supplementary Data 6 (sheet "Standard compounds") and referred to in the methods section (please see lines 493-494).

i) With regard to the stability of the samples for the NAD metabolome and the pre-analytical conditions, we have now added a section on "sample preparation" to the methods part.

j) SAM cycle measurements were obtained by LC-MS analysis, and we therefore refer to the response above. With regard to GSH measurements, the following information was provided by the NADMed company and, in a shortened version, has been added to the method description: "Stability of reduced and oxidized glutathiones is affected significantly by pH of the extraction buffer. Sulfosalicylic acid is usually used for protein precipitation in commercially available kits for glutathione measurement. It is not an optimal compound for maintaining stability of both reduced and oxidized glutathiones and can lead to underestimation of Glutathione content. NADMED extraction method which was used here for extraction of all four NADs and two glutathiones utilizes different principle. The extraction solution represents non-buffered complex mixture of organic solvents in water. The sample is injected into a pre-warmed extraction solution in order to

force protein unfolding and release of all non-covalently bound metabolites into solution. As there are no reducing agents, acid or base all pH and redox sensitive metabolites remain stable, including reduced and oxidized glutathione. As proteins are unfolded there is also no enzymatic activity, which could change concentration of interconvertible metabolites. During the next step proteins are removed. For this homogenate is quickly cooled down resulting in precipitation of proteins, which are then removed by centrifugation. Obtained supernatant (extract) is then used for separate measurement of glutathione pool (GSH +GSSG) and GSSG in enzymatic assay. In order to measure oxidized glutathione extract is treated with thiol scavenging reagent which efficiently masks free thiol group of GSH from being recognized in the assay. As a result, only GSSG is selectively measured. Next, GSSG value is subtracted from value of glutathione pool (GSH +GSSG) and concentration of reduced glutathione (GSH) is calculated.

Comment 5:

In the results in Fig 3 – what is NAD pool? In Fig 4 – how did the authors identify Me2PY? how certain that it is not Me-4 PY or a combination of these 2? Some metabolites have absolute quantification values expression in micro- / nano-molar, and other as relative intensity. This is not explained or justified....Validation of analytical methods when reporting quantitative values is essential, was not mentioned in this study.

Response:

a)The “NAD pool” is total NAD (NAD+ and NADH combined), this has been changed in figures and text for clarity.

b)The identification of Me2PY is based on accurate mass and retention time. According to MSOmics, in the applied method, there is a >1.5 min retention time difference between Me2PY and Me4PY. The retention times of the two compounds were obtained from running pure standards on the same setup.

c)For a few metabolites, only relative intensities were provided. In the case of homocysteine and NADP+, this is due to instability of the compound in the standard solutions. In the case of betaine and methionine, this is because those metabolites lacked a standard series for absolute quantification. This information has now been added to the methods section (please see lines 494497).

D) Regarding the stability of the compounds: sample analysis was started immediately after sample preparation without any intermediate storage. Therefore, any instability in the prepared samples would have been seen in the pooled QC samples that were analyzed repeatedly over the sequence. No clear trends of degradation were observed over the analytical sequence. The reported precision shows the standard deviation between these compounds in question including any possible degradation over the time in the autosampler. With regards to stability in the sample preparation

itself, this was not tested. However, samples were handled on ice or at 4°C during the sample preparation.

Information about the QC samples and the above-mentioned stability consideration have been added to the methods description (please see lines 466-469).

Comment 6:

6. In the discussion, the relationship between methyl pool (through measurement of SAM cycle intermediates) and NAD⁺ metabolome is not explored, even if these measurements were acquired...

Response:

The relationship between SAM and NAD is discussed in the Discussion on page 16, line 346. We are uncertain what additional information to add but would be happy to consider this at the editor's request.

Comment 7:

7. This is obviously a great deal of inter-individual variability in a clinical study. It is important to reflect and discuss this in the Discussion and Limitation sections. Is this high dose of NR really going to improve PD patients at a more general perspective beyond 20 subjects?

Response:

As mentioned also in response to the Reviewer's comment-3, the NR-SAFE study was not designed to established clinical efficacy, but to assess the short-term safety and tolerability of NR at a dose of 3000 mg per day, in order to enable further phase II/III trials to include and test this dose. As such the predefined outcomes of our trial were adequately assessed and fully met. The clinical efficacy of NR at this and other doses will be assessed by other, ongoing studies including the N-DOSE ([ClinicalTrials.gov](https://clinicaltrials.gov/ct2/show/study/NCT05589766) ID: NCT05589766) and NOPARK trials ([ClinicalTrials.gov](https://clinicaltrials.gov/ct2/show/study/NCT03568968) ID NCT03568968). This is clearly mentioned in the discussion. Additionally, to improved clarity, we have added the sentence "This study design was a phase I, single-center, randomized, double-blinded, placebo-controlled trial, aiming to provide initial evidence of safety and tolerability for NR at a dose of 3,000 mg daily" to the methods "participants and study design" section (please see lines 377-378).

#Additional changes

During the revision time, we became aware of a methodological artefact concerning our LC-MS measurements of the metabolite cyclic ADP ribose (cADPR). Due the nature of the MS method, the majority of the cADPR signal is derived from NAD by fractionation leading to removal of the nicotinamide ring and elimination of a water molecule, resulting in the same molecular mass as cADPR. Retention time analysis of our MS data confirmed this. Since we cannot confidently determine biological changes in cADPR levels, we have removed all information of cADPR from the manuscript. This is a very minor change which has no impact on any of the main conclusions of the work.

Due to journal formatting requirements, data and code availability have been separated into two separate sections. The section funding sources has been included in the acknowledgements section and ethics declarations has been included in the participants and study design section of the methods section.

Supplementary Tables 1-7 have been renamed Supplementary Data 1-7.

REVIEWERS' COMMENTS

Reviewer #1 (Remarks to the Author):

The authors have adequately addressed my concerns and have improved the quality of the manuscript.

Reviewer #4 (Remarks to the Author):

In their manuscript, Berven and colleagues report the results of a phase-1, randomized, placebo-controlled clinical trial on the safety and tolerability of high dose (1500 mg twice daily) nicotinamide riboside (NR)/placebo, administered for 30 days, in patients with Parkinson's disease. They found that patients with PD tolerated high dose of NR well.

The methodology is overall sound and the rationale for the need of an early-phase trial that investigates safety of high doses of NR in PD is convincing. The main objective of the trial is fulfilled, and the data support the authors' conclusion that the daily dose of NR can be expanded to 3000mg in phase II trials in PD, with appropriate monitoring.

The need for new treatments that mitigate PD progression is urgent and still unmet, which requires reconsidering the design of clinical trials, towards less expensive and time-consuming alternatives. Approaches that include repurposing of existing agents, biomarkers measuring treatment response on the tissue level and specific measures of clinical progression are of great value. The present study serves this purpose well, is well written and provides adequate context to the reported results.

Please find here my comments on some parts of the methodology I considered important to look closer to.

- Figure 1: Are there more details available on the "other reasons" one patient was excluded at screening?
- Female/male ratio in the study is 1:3, which is way below the 1:1.4 (PMID: 30287051) that is considered representative for PD population. This needs to be addressed in the Limitations. Sex bias in clinical trials is an important issue (PMID: 37385457; PMID: 37041049) worth attention.
- Table 1: NR-group were younger and had lower BMI than placebo-group and had fewer years since diagnosis. These parameters did not show statistically significant differences, but groups are small and there was an obvious, and maybe biologically relevant difference between groups. This maybe should be kept in mind in the way results are interpreted.
- Line 111: "Hoehn and Yahr stage is an established scale for PD progression". I do not agree that H&Y is a scale measuring PD progression, but a rough tool measuring specific aspects of motor symptom severity. I think it would be more informative to show comparisons of the following parameters to support the groups' comparability: cognitive scores (e.g. MoCA), Schwab and England score of functional dependence (number of participants with score <80%) and number of participants with H&Y>2 (instead

of mean scores).

- I have some concerns regarding the validity of the authors' conclusion regarding the claim that "High dose NR is associated with clinical improvement". Firstly, clinical improvement during such a short treatment period would probably mean that NR has an immediate, symptomatic effect. In my understanding this is not what is expected by an agent assumed to increase neuronal resilience and delay neurodegeneration, which would presumably show its positive impact on MDS-UPDRS score at a later timepoint.

Secondly, and most importantly, MDS-UPDRS in ON state is not a great tool to measure differences in clinical severity during so short time, but also otherwise. Several parameters can affect the result of the score in ON state, intra-individual variability can be high, LEDD score needs to be taken into consideration, the rater(s) need to be fixed along the study, as well as the time of the day when the patient-visits occurred. The fact that time since last levodopa-dose differed, also after the authors' try to match more comparable groups, makes the conclusion even weaker.

With that said, the study is not powered to show differences in this outcome measure (MDS-UPDRS score/sub-scores) and it is also beyond the main objective of the study, thus I think it would be enough to further soften this conclusion, rephrase and take these aspects into consideration.

The duration of the study is short to be able to draw conclusion on long-term toxicity and late adverse events, however the trial was well designed to assess short-term safety. The authors seem to have a well-planned project pipeline for the investigation of the NR efficacy in PD, including another two ongoing, randomized trials that will hopefully shed more light in the major question of long-term efficacy vs. toxicity.

Regarding power analysis for the primary objective of the trial, in my opinion, it is acceptable to report AEs using a qualitative, descriptive approach, given the short duration and early phase of the trial.

A minor point for discussion, rather than critique, is that the terms "neuroprotective agents" and "disease-modifying therapies" may be counterproductive in the joint efforts towards the ultimate goal of developing effective therapies against PD, for reasons nicely explained in this paper: PMID: 33026697. I thought the authors may be interested to reflect on that in their current and future work.

Reviewer #5 (Remarks to the Author):

Title: Supplementary_Data_6_metabolomics

Reviewer #3

Comments on Authors' response for Reviewer #3

- Authors have addressed the reviewer #3's points 1-2
- Comment 3: Agree with the reviewer, the sample size is too small (n=20) to determine pharmacological dose response of NR which is tested at high dose of (3,000 mg/day) nicotinamide riboside for 4 weeks in very small sample size of 20 people (n=10/group) with Parkinson's disease.
- Comment 4: Part of the points raised by reviewer have been addressed. It seems NADMED is the company who did the LCMS measurements of NAD⁺/NADH. LCMS platform should be able to measure NADP⁺ etc metabolites in the sample. The further information about the methodology could not be located in the supplementary table or in the methodology.
- It is not so clear whether authors used single method for oxidized and reduced nucleotide panel or two independent LCMS/MS methods. Further clarity on the method section is desired.
- Supplementary Table 5, sheet 2: NADP⁺ level looks very high than NADPH. Our conventional wisdom and most of the literature teach us NADPH pool is greater than NADP⁺. There is high likely NADPH is oxidized easily to NADP⁺ if the pH of the extraction solvent is highly acidic which might introduce artifacts for NADP measurement. Authors need to reconsider checking the NADP⁺ level in the sample. In the supplementary table 6, sheet 2 & 7, authors have provided AUCs only for NADP⁺! why the inconsistencies in the report?
- Supplementary Table 6 sheet 2 & 7: Reference to whole blood/ urine data shows higher dose of NR is associated with increased level of NAD⁺, nicotinamide and methyl nicotinamide in blood (sheet 3) and urine (sheet 7) after 4 weeks of NR supplementation compared to baseline. Elevated levels of nicotinamide and methyl nicotinamide could be metabolites derived from NR deribosylation and absorbed in the body as nicotinamide followed by its conversion to NAD⁺. Higher level of NR in urine than whole blood raises the question about the bioavailability of intact NR and its absorption. There is limited evidence of absorption of NR as intact molecule in the human body and about its bioavailability. Therefore, further studies are needed with stable isotope labelled NR specially in human. Chellappa et al., 2022 from Baur lab showed nicely the fate of stable isotope labelled NR in mice and is converted to nicotinamide and the labelled nicotinamide was detected in NAD⁺ in various tissues in mice. The bioavailability of NR in whole blood is low in nM range so despite the increased dose of NR (3000 mg/day) the systemic NR level did not increase in whole blood rather excreted through urine. The bioavailability of intact NR, at which is the extent and rate enters the bloodstream and reaches the target tissues, is not well established. Therefore, further research is required to measure and compare the pharmacokinetic profiles of NR and its metabolites in different formulations and dosages. More research is needed to determine how NR is absorbed in vivo, its optimal dose and safety of NR supplementation in humans.

Point-by-point reply to the Reviewers' Comments:

Reviewer #4 (Remarks to the Author):

In their manuscript, Berven and colleagues report the results of a phase-1, randomized, placebo-controlled clinical trial on the safety and tolerability of high dose (1500 mg twice daily) nicotinamide riboside (NR)/placebo, administered for 30 days, in patients with Parkinson's disease. They found that patients with PD tolerated high dose of NR well.

The methodology is overall sound and the rationale for the need of an early-phase trial that investigates safety of high doses of NR in PD is convincing. The main objective of the trial is fulfilled, and the data support the authors' conclusion that the daily dose of NR can be expanded to 3000mg in phase II trials in PD, with appropriate monitoring.

The need for new treatments that mitigate PD progression is urgent and still unmet, which requires reconsidering the design of clinical trials, towards less expensive and time-consuming alternatives. Approaches that include repurposing of existing agents, biomarkers measuring treatment response on the tissue level and specific measures of clinical progression are of great value. The present study serves this purpose well, is well written and provides adequate context to the reported results.

Please find here my comments on some parts of the methodology I considered important to look closer to.

Comment 1:

Figure 1: Are there more details available on the "other reasons" one patient was excluded at screening?

Response:

We thank the reviewer for the comment. The participant decided to withdraw from the study due to personal time constraints and, therefore, withdrew consent. This has been adjusted to "withdrew consent" in Figure 1.

Comment 2:

Female/male ratio in the study is 1:3, which is way below the 1:1.4 (PMID: 30287051) that is considered representative for PD population. This needs to be addressed in the Limitations. Sex bias in clinical trials is an important issue (PMID: 37385457; PMID: 37041049) worth attention.

Response:

We thank the reviewer for this comment. We have included this under the limitations section of the manuscript at lines 336-343.

Comment 3:

Table 1: NR-group were younger and had lower BMI than placebo-group and had fewer years since diagnosis. These parameters did not show statistically significant differences, but groups are small and there was an obvious, and maybe biologically relevant difference between groups. This maybe should be kept in mind in the way results are interpreted.

Response:

We thank the reviewer for this observation and have included this information in the results section at lines 110-113.

Comment 4:

Line 111: "Hoehn and Yahr stage is an established scale of PD progression". I do not agree that H&Y is a scale measuring PD progression, but a rough tool measuring specific aspects of motor symptom severity. I think it would be more informative to show comparisons of the following parameters to support the groups' comparability: cognitive scores (e.g. MoCA), Schwab and England score of functional dependence (number of participants with score <80%) and number of participants with H&Y>2 (instead of mean scores).

Response:

We thank the reviewer for the comments. We have removed the sentence about the H&Y scale being a scale of PD progression on lines 113-115. MoCA and S&E were not carried out and we are therefore not able to include these in the manuscript. We have added the number of participants with H&Y score above 2 to Table 1, and have removed the mean Hoehn and Yahr scores from this table. Below is a cut-out showing the changes made to Table 1.

Comment 5:

I have some concerns regarding the validity of the authors' conclusion regarding the claim that "High dose NR is associated with clinical improvement".

Firstly, clinical improvement during such a short treatment period would probably mean that NR has an immediate, symptomatic effect. In my understanding this is not what is expected by an agent assumed to increase neuronal resilience and delay neurodegeneration, which would presumably show its positive impact on MDS-UPDRS score at a later timepoint.

Secondly, and most importantly, MDS-UPDRS in ON state is not a great tool to measure differences in clinical severity during so short time, but also otherwise. Several parameters can affect the result of the score in ON state, intra-individual variability can be high, LEDD score needs to be taken into consideration, the rater(s) need to be fixed along the study, as well as the time of the day when the patient-visits occurred. The fact that time since last levodopa-dose

differed, also after the authors' try to match more comparable groups, makes the conclusion even weaker.

With that said, the study is not powered to show differences in this outcome measure (MDS-UPDRS score/sub-scores) and it is also beyond the main objective of the study, thus I think it would be enough to further soften this conclusion, rephrase and take these aspects into consideration.

Response:

In response to the first point, we respectfully disagree that the possibility of a symptomatic NR effect can be excluded. The effects of high dose NR on PD are currently unknown. For instance, it is not unthinkable that neuronal NAD-augmentation could ameliorate the metabolism and improve the function of surviving dopaminergic neurons, thereby resulting in higher dopaminergic transmission. A slight symptomatic effect has also been observed in the NADPARK study, where 1000 mg NR for 30 days was associated with a mild MDS-UPDRS improvement in individuals showing a strong NAD augmentation as response to the treatment (Brakedal et al 2022). Having said that, we fully agree with the reviewer's comment regarding the effect on neuronal resilience and delay of neurodegeneration, as 4 weeks is too short time course to observe such effects.

In response to the second point, we agree with the reviewer and have significantly toned down the interpretation of this observation, both in the Results section and in the Discussion (lines 185-186 and 266-275, respectively).

Comment 6:

The duration of the study is short to be able to draw conclusion on long-term toxicity and late adverse events, however the trial was well designed to assess short-term safety. The authors seem to have a well-planned project pipeline for the investigation of the NR efficacy in PD, including another two ongoing, randomized trials that will hopefully shed more light in the major question of long-term efficacy vs. toxicity.

Response:

We thank the reviewer for the comments and the assessment of our work.

Comment 7:

Regarding power analysis for the primary objective of the trial, in my opinion, it is acceptable to report AEs using a qualitative, descriptive approach, given the short duration and early phase of the trial.

Response:

We thank the reviewer for this comment.

Comment 8:

A minor point for discussion, rather than critique, is that the terms “neuroprotective agents” and “disease-modifying therapies” may be counterproductive in the joint efforts towards the ultimate goal of developing effective therapies against PD, for reasons nicely explained in this paper: PMID: 33026697. I thought the authors may be interested to reflect on that in their current and future work.

Response:

We thank the reviewer for this comment and will reflect on this further during our work. For now, we respectfully choose to keep the formulations as presented in the manuscript and look forward to further discussions on this topic.

Reviewer #5:**Comment 1:**

Authors have addressed the reviewer #3's points 1-2

Response:**Comment 2:**

Comment 3 (of Reviewer #3): Agree with the reviewer, the sample size is too small (n=20) to determine pharmacological dose response of NR which is tested at high dose of (3,000 mg/day) nicotinamide riboside for 4 weeks in very small sample size of 20 people (n=10/group) with Parkinson's disease.

Response:

We thank the reviewer for this comment. As we state in the manuscript, the primary objective of this study was to assess the short-term safety of this high dose NR treatment, in order to establish whether it is clinically acceptable to employ this dose in further phase II trials. We did not attempt to study pharmacological dose responses. Our study is of sufficient size to address our predetermined end points, and of a size typical for safety trials.

We are also aware that all observations beyond this, including the clinical improvement observed in the NR group, are limited by the size of the study, and may be influenced by other factors as well. This is referred to in both the results and discussion section of the manuscript.

Comment 3:

Comment 4 (of Reviewer #3): Part of the points raised by reviewer have been addressed. It seems NADMED is the company who did the LCMS measurements of NAD⁺/NADH. LCMS platform should be able to measure NADP⁺ etc metabolites in the sample. The further information about the methodology could not be located in the supplementary table or in the methodology.

Response: We thank the reviewer for this comment. In the NR-SAFE study, we have carried out two different methods for metabolite detection: 1) an LC-MS based approach detecting a panel of NAD-related metabolites and other metabolites (carried out by MS-Omics, Denmark), and 2) the NADMed assay (carried out by the NADMed company, Finland (www.nadmed.fi) which is a spectrophotometric analysis of NAD⁺, NADH, NADP⁺, NADPH, GSH and GSSG. The method description can be found in the method section, and the results are separately presented: Data derived from NADMed analysis are presented in Figure 3 and Supplementary Data 5, data from the LC-MS analysis are presented in Figure 4 and Supplementary Data 6.

While we do report NADP⁺ and NADPH data from the NADMed analysis, the applied LC-MS methods were only able to detect NADPH in the samples above the limit of detection (LOD). This is indicated in Supplementary Data 6, sheet “whole blood all data”. Values from all samples are shown also for NADPH, but most are below LOD. We also added a sentence to the method section to further emphasize this. Thus, calculations as shown in sheets “Baseline whole blood NR vs PL” and “Analyses LCMS whole blood” were not possible. Values for NADP⁺ are reported also in the LCMS analysis.

Comment 4:

It is not so clear whether authors used single method for oxidized and reduced nucleotide panel or two independent LCMS/MS methods. Further clarity on the method section is desired.

Response:

We thank the reviewer for this comment. In the LC-MS analysis, two extraction methods were applied to blood samples, a polar and a semi-polar extraction method. Which method is used for the reported values is indicated in Supplementary Data 6 (this is also referred to in the method description). As also indicated in Response to comment 3 (see above) NAD⁺, NADH, NADP⁺, NADPH, GSH and GSSG were also analyzed by the spectrophotometric NADMed assay. Here, the method description made available by the company and included in our method section states that: “...each metabolite was measured individually using optimized cyclic enzymatic assays with colorimetric detection”.

We hope this further clarifies the methodology and resolves the concerns of the reviewer.

Comment 5:

Supplementary Table 5, sheet 2: NADP⁺ level looks very high than NADPH. Our conventional wisdom and most of the literature teach us NADPH pool is greater than NADP⁺. There is high likely NADPH is oxidized easily to NADP⁺ if the pH of the extraction solvent is highly acidic which might introduce artifacts for NADP measurement. Authors need to reconsider checking the NADP⁺ level in the sample. In the supplementary table 6, sheet 2 & 7, authors have provided AUCs only for NADP⁺! why the inconsistencies in the report?

Response:

We thank the reviewer for this comment and agree that most reports indicate that the **intracellular** NADP⁺/NADPH ratio lies on the side of the reduced form. While we cannot fully rule out an oxidation of some of the metabolites during the sample processing, it should be pointed out that the samples are analyzed together with a standard curve of known

concentrations that undergoes the same extraction procedure, so all artefactual changes should be similar for both standards and samples. Moreover, we show intra-individual changes from before and after treatment, and observe that the NADP⁺ and total NADP levels increase, but not NADPH or the redox ratio (at least not significantly). Any effect of the extraction method should affect samples from before and after treatment similarly. Finally, the results of an increased NADP⁺ level are confirmed by two independent methods (NADMed and LC-MS). However, since we cannot rule out the possibility of the methodology affecting the redox status, we also report the total NADP content, which should be unaffected by redox changes.

In Supplementary data 6, the NADP⁺ values (as well as the values of a few other metabolites) from the LC-MS analysis are reported as relative values of AUC (area under the curve).

In our method description, the explanation for this reads as follows: “For a few metabolites only relative intensities (areas) are provided. In the case of homocysteine and NADP⁺, this is due to instability of the compound in the standard solutions.”

Since NADP⁺ was readily detected in the samples, we provide these values for relative comparison, even though we cannot calculate the absolute concentrations in the samples.

As mentioned above, values for NADPH are not shown because it was detected below the “limit of detection” of the applied LC-MS method.

We hope this clarifies this question and resolves the concern of the reviewer.

Comment 6:

Supplementary Table 6 sheet 2 & 7: Reference to whole blood/ urine data shows higher dose of NR is associated with increased level of NAD⁺, nicotinamide and methyl nicotinamide in blood (sheet 3) and urine (sheet 7) after 4 weeks of NR supplementation compared to baseline. Elevated levels of nicotinamide and methyl nicotinamide could be metabolites derived from NR deribosylation and absorbed in the body as nicotinamide followed by its conversion to NAD⁺. Higher level of NR in urine than whole blood raises the question about the bioavailability of intact NR and its absorption. There is limited evidence of absorption of NR as intact molecule in the human body and about its bioavailability. Therefore, further studies are needed with stable isotope labelled NR specially in human. Chellappa et al., 2022 from Baur lab showed nicely the fate of stable isotope labelled NR in mice and is converted to nicotinamide and the labelled nicotinamide was detected in NAD⁺ in various tissues in mice. The bioavailability of NR in whole blood is low in nM range so despite the increased dose of NR (3000 mg/day) the systemic NR level did not increase in whole blood rather excreted through urine. The bioavailability of intact NR, at which is the extent and rate enters the bloodstream and reaches the target tissues, is not well established. Therefore, further research is required to measure and compare the pharmacokinetic profiles of NR and its metabolites in different formulations and dosages. More research is needed to determine how NR is absorbed in vivo, its optimal dose and safety of NR supplementation in humans.

Response:

We thank the reviewer for this comment and the comprehensive assessment of the knowledge status around NR supplementation in humans. We fully agree that there is little known about

the bioavailability of intact NR; to which extent and at which rate it enters the bloodstream (if at all). The involvement of the microbiome is probably the most exciting factor of this equation, and several excellent articles have supported the idea that NR (and other NAD precursors) are made available to human use by preparations carried out by the microbiome(both via generation of nicotinamide, but also deamidation to nicotinic acid (NA), and utilization of NA in the deamidated pathway to NAD (which is supported for example by the continued detection of NAAD, an intermediate of the deamidated NAD biosynthetic pathway that is not generated in the “direct route” from NR to NAD, in human studies of NR supplementation.

We also fully agree that more research is needed to determine how NR is used in the human body to support NAD synthesis, its optimal dose and not least safety of NR supplementation. We therefore take this comment as an encouragement for our future studies that will investigate some of these exact points and looking forward to continued discussion and discoveries in the field.

Additional Comments:

Due to data privacy laws, individual data on age and sex has been removed, as this could theoretically be used to identify participants. Data are presented as aggregate data. The relevant data has been removed in the Supplementary Data files and Source Data file.

The number of participants used in the initial analysis of “time since levodopa” has been changed to $n = 9$ in each group, as one participant in each group did not take levodopa and, thus, time since levodopa was not registered. This has been commented on in the legends of Table 4 and Supplementary Data 4.